# Aging impairs the antiviral defense in *Caenorhabditis elegans* due to loss of DRH-1/RIG-I deSUMOylation by ULP-4/SENP7

Yun Zhang & Andrew V Samuelson ⬚ ✉

## Abstract

Innate immune defense relies on post-translational modifications (PTMs) to protect against viral infections. SUMOylation plays complex roles in viral replication and antiviral defenses in mammals and has been implicated in age-associated diseases. Whether PTMs and SUMOylation contribute to age-induced immunosenescence is unknown. We find that antiviral defense in *Caenorhabditis elegans* is regulated through SUMOylation of DRH-1, ortholog of the cytosolic pattern recognition receptor RIG-I. The SUMO isopeptidase ULP-4 is essential for deSUMOylation of DRH-1 and activation of the intracellular pathogen response (IPR) after exposure to Orsay virus (OV). ULP-4 stabilizes DRH-1, which translocates to the mitochondria to activate the IPR. Loss of *drh-1* or *ulp-4* compromises antiviral defense; mutant animals fail to clear OV and develop intestinal pathogenesis. During aging, *ulp-4* expression decreases, which promotes DRH-1 proteosomal degradation and IPR loss. Mutating the DRH-1 SUMOylated lysines partially rescued the age-associated lost inducibility of the IPR. Our work establishes that aging results in dysregulated SUMOylation and loss of DRH-1, which compromises antiviral defense and creates a physiological shift to favor chronic pathological infection in older animals.

**Keywords** *C. elegans*; Orsay Virus; Antiviral Defense; Aging; SUMOylation
**Subject Categories** Microbiology, Virology & Host Pathogen Interaction; Molecular Biology of Disease; Post-translational Modifications & Proteolysis

## Introduction

Viral infection is universal to all organisms. However, the physiological state of the host plays a major role in shaping host–viral interactions and the outcome of an infection. Numerous factors influencing an organism's physiological status: environmental conditions, diet and nutrition, sleep, comorbidities, genetic polymorphisms, and aging can all influence the severity and outcome of an acute infection (Casadevall and Pirofski, 2018). For instance, the COVID-19 pandemic highlighted the elderly as being especially vulnerable to increased disease severity and mortality, as well as an increased likelihood of long-term complications (Farshbafnadi et al, 2021). In contrast, young individuals were often asymptomatic or had relatively mild symptoms, and long-term complications were rare (Farshbafnadi et al, 2021). While some hallmarks of aging are known to increase vulnerability to infection, for example, immunosenescence, how the physiological state of "being old" determines the outcome of viral infections remains poorly understood.

The innate immune response is the first line of defense against invading viruses. In mammals, infected cells initiate a rapid innate immune response by detecting the DNA or RNA of the invading viruses with pathogen pattern recognition receptors (PRR), including the Toll-like receptors (TLR), retinoic acid-inducible gene I (RIG-I) like receptors (RLR), the cyclic GMP-AMP (cGAMP) synthase (cGAS), and nucleotide-binding oligomerization domain (NOD)-like receptors (NLR) (Ma and Damania, 2016). PRRs trigger the production of immune effectors, which include cytokines and chemokines (Salazar-Mather and Hokeness, 2006). Among these immune effectors, the Type I interferon response plays a crucial role in regulating antiviral innate immune response. One important pathway that triggers the production of type I interferons (IFN-I) by RLRs includes RIG-I and melanoma differentiation-associated gene 5 (MDA5). Once activated, RIG-I and MDA5 translocate to the outer membrane of mitochondria and bind to mitochondrial antiviral-signaling proteins (MAVS) to initiate a signaling cascade that facilitates cell death and removal of the infected cells (Kell and Gale, 2015).

*Caenorhabditis elegans* infection by Orsay virus (OV), a natural pathogen, is a novel and powerful genetic system to elucidate how aging impacts host–viral interactions, physiological response, and disease outcome. Mammalian cell culture lacks systemic-level responses seen in vivo and can lead to in vitro artifacts (Nikolich-Zugich, 2018). Mechanisms preserving cell function at the molecular/subcellular level are difficult to discover in vivo within higher metazoans, as they are often masked by cell death/replacement and immune clearance mechanisms that maintain tissue homeostasis; these issues are further confounded by compensatory and redundant mechanisms, which could mask the genetic contribution of underlying deficiencies. In contrast, the

Department of Biomedical Genetics, University of Rochester Medical Center, Rochester, NY, USA. ✉E-mail: Andrew_Samuelson@URMC.Rochester.edu

absence of an acquired immune system in *C. elegans*, combined with an innate immune system that does not rely upon cell death or dedicated immune cells, allows investigation into how non-immune cells maintain homeostasis in vivo. Approximately eight percent of the human genome consists of remnants of viral sequence, some of which are still widely expressed throughout normal tissues (Burn et al, 2022), a testament that viral infection does not always result in cell death. *C. elegans* has provided key insight into biological principles of aging (Lazaro-Pena et al, 2022; Zhang et al, 2020), and provides a powerful, underutilized in vivo system to explore how aging alters antiviral defense in normal cells and tissues.

OV is a small, enteric, single-stranded RNA+ virus of the nodavirus family and the only known natural viral pathogen of *C. elegans* (Felix et al, 2011). Infection activates DRH-1, the ortholog of the DEAD/H-box Helicase and cytosolic pattern recognition receptor RIG-I to limit viral infection and pathogenesis (Ashe et al, 2013; Batachari et al, 2024; Coffman et al, 2017; Gammon et al, 2017; Guo et al, 2013; Lu et al, 2009; Sowa et al, 2020). The related Santeuil, and Le Blanc viruses infect *Caenorhabditis briggsae* to induce an evolutionarily conserved response (Chen et al, 2017; Franz et al, 2014); *C. elegans* and *C. briggsae* diverged between 30 and 100 million years ago, the further estimate is on par with the evolutionary distance between mice and humans (Cutter, 2008; Ernst and Carvunis, 2018; Gupta et al, 2007; Lucas et al, 2018; Pervouchine et al, 2015). The OV genome is segmented into two parts: RNA1 encodes an RNA-dependent RNA polymerase (RdRP), and RNA2 encodes capsid and delta proteins (Guo et al, 2020; Jiang et al, 2014; Yuan et al, 2018). The *C. elegans* intestine consists of 20 non-renewable cells, which last the lifespan of the animal. OV typically only infects between two to six intestinal cells; loss of RNAi has been reported to increase the amount of viral protein within a cell without increasing the number of infected cells (Franz et al, 2014), which suggests RNAi acts solely within the infected cell to reduce levels of virus. In a separate pathway that is independent of RNAi, DRH-1 also activates ZIP-1 (a bHLH transcription factor) to induce the Intracellular Pathogen Response (IPR) (Lazetic et al, 2022; Reddy et al, 2017). The IPR constitutes a shared adaptive transcriptional response of ~80 genes induced by OV, fungal pathogens, and some non-pathogenic forms of stress, including proteotoxicity following prolonged heat stress or inhibition of the proteosome (Lazetic et al, 2023; Lazetic et al, 2022). The IPR is distinct from canonical proteostasis pathways; mechanistically, the IPR includes components of the CUL-6/cullin-ring ubiquitin ligase complex, and regulators of the IPR have similarities to the Type I IFN response (Bardan Sarmiento et al, 2024; Lazetic et al, 2023; Panek et al, 2020).

In the current study, we sought to gain mechanistic insight into how DRH-1 is activated by viral infection and how the antiviral activity of DRH-1 changes during aging. We found that SUMOylation of DRH-1 negatively regulated the IPR and facilitated viral infection. SUMOylation is a rapid and reversible post-translational modification that plays conflicting roles in viral replication and antiviral defenses in mammals (Fan et al, 2022). We found increased viral transcript levels, bloating within the intestinal lumen, failure to clear the virus, and impaired induction of the IPR in animals lacking the SUMO isopeptidase *ulp-4* (orthologous to mammalian SENP7). In the absence of *ulp-4*, basal levels of DRH-1 are reduced in part via increased degradation through the proteosome, consistent with findings that SUMOylated proteins

can be degraded by the ubiquitin-proteasome system (Marco et al, 2015). After infection, ULP-4 deSUMOylated DRH-1 to facilitate translocation to the mitochondrial outer membrane to activate the IPR; mutating DRH-1 lysine residues that are targets for SUMOylation rescued the impaired IPR and levels of DRH-1 protein. During aging, *ulp-4* mRNA decreased, and the inducibility of the IPR was lost. Overexpression of a non-SUMOylatable isoform of DRH-1, partially restored the inducibility of the IPR in older animals, which suggests that age-associated dysregulation of DRH-1 SUMOylation contributes to an increase in pathogenesis in older animals.

## Results

### The SUMO isopeptidase ULP-4 is required for the induction of the intracellular pathogen response by viral infection

SUMOylation is a post-translational modification process resembling ubiquitination; the *C. elegans* genome encodes a single SUMO moiety (SMO-1), two E1 ligases (UBA-2, AOS-1), one E2 ligase (UBC-9) and one E3 enzyme (GEI-17), which conjugate SMO-1 onto proteins. Conversely, there are four SUMO isopeptidases that deconjugate SUMO from a target protein: ULP-1, ULP-2, ULP-4, and ULP-5. To begin to explore the roles of SUMOylation in the regulation of antiviral innate immunity, we inactivated components of the SUMOylation machinery by feeding RNAi and assessed the impact on the induction of the IPR after viral infection. Induction of *pals-5* expression (*pals-5::GFP*) is a canonical readout for induction of the IPR (Jiang et al, 2017; Reddy et al, 2017; Sowa et al, 2020). Of the four SUMO isopeptidases, only loss of *ulp-4* significantly limited *pals-5p::GFP* induction after viral infection (Fig. 1A; Figure EV1A), which was approximately 4-fold lower than induction in virally infected animals treated with empty vector RNAi (Fig. 1B). Conversely, inactivation of components that promote SUMO conjugation (*aos-1(RNAi), gei-17(RNAi), ubc-9(RNAi)*), or loss of SUMO itself (*smo-1(RNAi)*), significantly induced *pals-5p::GFP* after infection (Fig. 1B; Figure EV1A). Inactivation of any component of the SUMO core machinery was not sufficient to induce *pals-5* in the absence of viral infection (Fig. 1A and Fig. EV1A), suggesting that altered SUMOylation status plays a regulatory role in the response to viral infection. To confirm that loss of SUMO conjugation enhanced induction of the IPR, we tested whether *gei-17(tm2723)* deletion mutants recapitulated *gei-17(RNAi)*; viral infection of *gei-17* mutant animals induced *pals-5p::GFP* approximately 3-fold higher than induction in wild-type animals (Fig. EV1B,C). Next, we tested whether *ulp-4(tm3688)* deletion mutants recapitulated *ulp-4(RNAi)*; the relative induction of *pals-5* mRNA was approximately fivefold less in the absence of *ulp-4* (Fig. 1C). Viral load was higher in the absence of *ulp-4*, either via RNAi or in null-mutant animals (Fig. 1D,E). Next, we tested whether non-viral triggers that induce the IPR also required *ulp-4* (i.e., proteasome blockade or prolonged heat stress (Sowa et al, 2020); *pals-5p::GFP*). Significantly, we found *ulp-4* is not required for the induction of *pals-5* by these two non-viral triggers (Appendix Fig. S1). We conclude that ULP-4 has an essential and specific role in activating an antiviral defense response after OV infection.

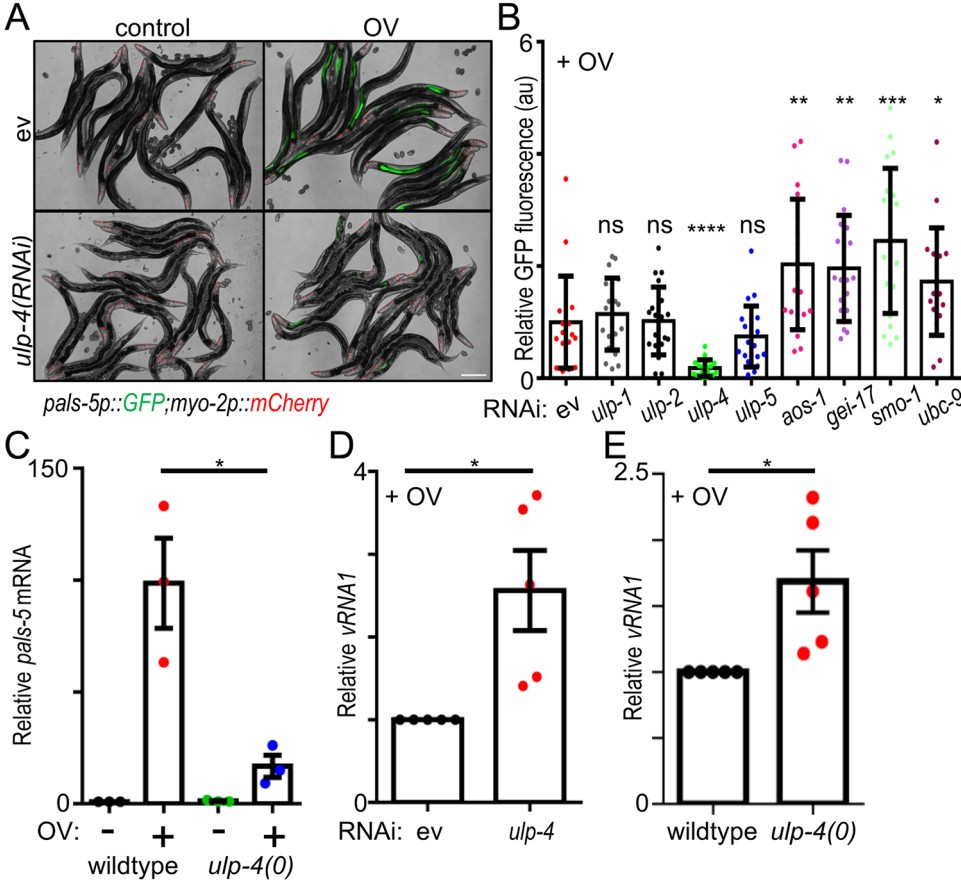

**Figure 1. ULP-4 is required for the induction of the intracellular pathogen response after viral infection.**

(A) Representative images of *pals-5p::GFP* expression after empty vector (ev) or *ulp-4* RNAi +/− viral infection. *myo-2p::mCherry* is a pharyngeal-expressed co-injection marker to identify the *jyIs8[pals-5p::gfp]* transgene. Scale bar = 200 μm. Images are reused in Fig. EV1A, which includes an expansion of panels covering additional conditions that were tested in the same experiment. (B) Quantification of *pals-5p::GFP* fluorescence intensity after inactivation of SUMOylation machinery + viral infection. Representative images +/− viral infection are in (A) and Fig. EV1A. *P = 0.0144, **P = 0.0025 (aos-1) P = 0.0015 (gei-17), ***P = 0.001, ****P < 0.0001, ns (no significant difference): P = 0.5406 (ulp-1), P = 0.9302 (ulp-2), and P = 0.2551 (ulp-5). (C) RT-qPCR analysis of endogenous *pals-5* mRNA in either wildtype or *ulp-4(tm3688)* null mutant animals +/− viral infection. *P = 0.0169. (D) RT-qPCR analysis of Orsay virus *RNA1* levels in (A). * P = 0.0112. (E) RT-qPCR analysis of Orsay virus *RNA1* levels in (C). *P = 0.0196. In all cases, the relative mean of 20 animals per trial, across three independent biological trials, is shown. Error bars are the SEM. A two-tailed *t*-test was used to calculate the indicated P values. Source data are available online for this figure.

## ULP-4 acts in the intestine to regulate antiviral immune response

To explore whether ULP-4 regulates antiviral immune signaling cell autonomously within the intestine or cell non-autonomously, we used tissue-specific promoters to rescue *ulp-4* expression with the intestine (*ges-1p*), muscle (*myo-3p*), epidermis (*dpy-7p*), or nervous system (*rab-3p*) in the *ulp-4(tm3688)* null mutants, and then assessed whether *pals-5* induction was restored following viral infection. We found that only intestinal *ulp-4* expression (*ges-1p::ulp-4*) restored *pals-5* induction after viral infection, as measured by GFP fluorescence (*pals-5p::GFP*) and levels of endogenous mRNA (Figs. 2A,B and EV2). Next, we conducted the converse experiment using intestinal-specific RNAi. Intestinal loss of *ulp-4* impaired the inducibility of the IPR to a comparable extent as systemic inactivation (Fig. 2C). We conclude that ULP-4 functions within the intestine to regulate induction of the IPR in response to viral infection, which is consistent with either a cell autonomous or localized intestinal response to an enteric viral infection.

## ULP-4 acts in a common pathway with DRH-1 to limit viral infection

To begin to understand how ULP-4 activity regulates the IPR in response to viral infection in young animals, we first determined whether ULP-4 and DRH-1 acted within a common genetic pathway. Consistent with previous reports that DRH-1 signaling is necessary for the induction of the IPR (Sowa et al, 2020), we found that *drh-1(ok3495)* null mutant animals failed to induce the IPR following viral infection (*pals-5p::GFP*; Fig. 3A). Next, we found that simultaneous loss of both *ulp-4* and *drh-1* did not produce higher viral loads in infected animals: inactivating *ulp-4* in otherwise wildtype animals resulted in a small but significant increase in viral loads, as measured by levels of Orsay virus *vRNA1* (Fig. 3B). In contrast, infected *drh-1(ok3495)* null mutant animals

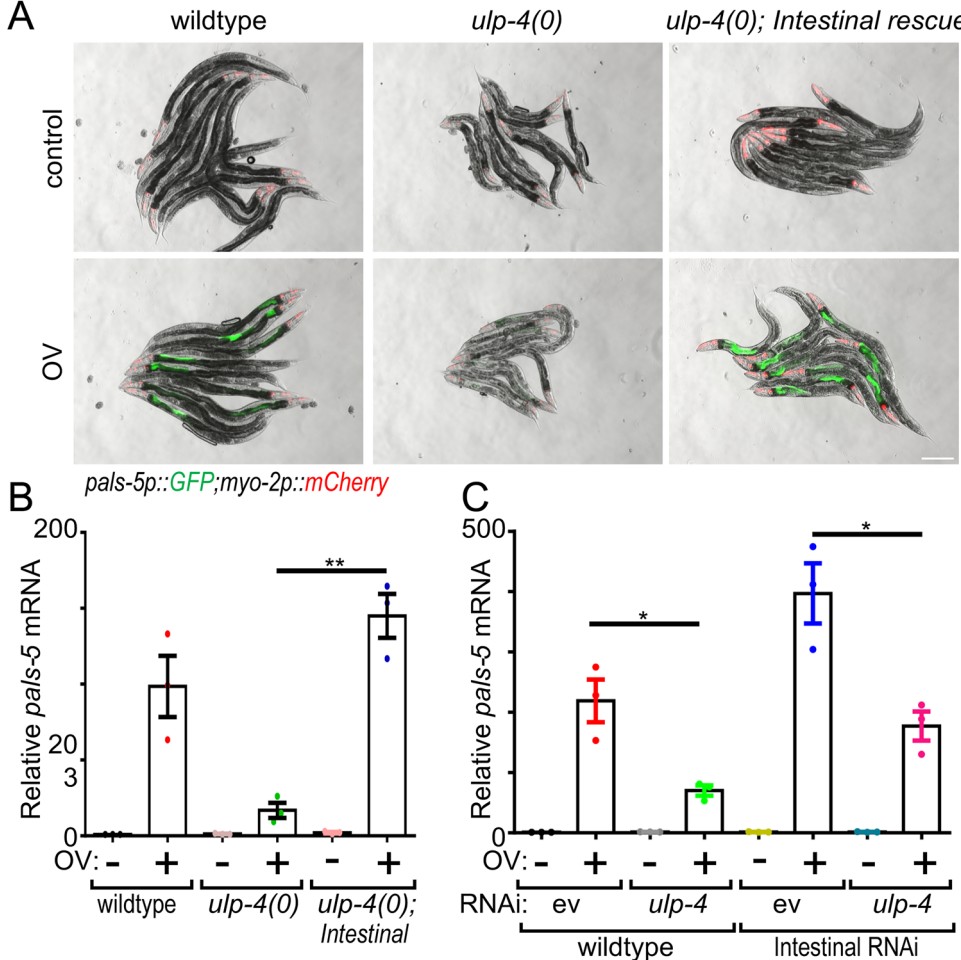

**Figure 2. ULP-4 acts in the intestine to regulate the IPR upon viral infection.**

(A) Representative images of *pals-5p::GFP* expression +/− viral infection of either wildtype, *ulp-4(tm3688)* null, or *ulp-4(tm3688);artEx95(ges-1p::ULP-4)* animals. Scale bar = 200 µm. (B) RT-qPCR analysis of endogenous *pals-5* mRNA of samples in (A). **P = 0.0011. (C) RT-qPCR analysis of endogenous *pals-5* mRNA +/− viral infection after systemic or intestine-specific inactivation of *ulp-4*. *P = 0.0151 (*ulp-4(RNAi)* in wild-type) and P = 0.0165 (*ulp-4(RNAi)* in animals with RNAi restricted to intestine). In all cases, values are the mean across three independent biological trials; error bars are SEM. A two-tailed *t*-test was used to calculate P values. Source data are available online for this figure.

had viral loads ~8.4-fold higher than wildtype; there was no significant additivity when both *ulp-4* and *drh-1* were lost (Fig. 3B). It is unlikely that a lack of additivity was due to a threshold effect, as impairing other antiviral innate defenses, such as RNAi, can produce viral loads several orders of magnitude greater than those observed in wildtype (Felix et al, 2011). The relative increase in viral loads were negatively correlated with the level of IPR induction after loss of either *ulp-4* or *drh-1* (compare Figs. 2C and 3A,B for loss of *ulp-4* and *drh-1*, respectively). This suggests that while ULP-4 and DRH-1 have common antiviral functions, in the absence of *ulp-4*, DRH-1 retains some antiviral activity.

## ULP-4 prevents proteosomal degradation of DRH-1

We tested whether modulation of SUMOylation in vivo would alter the stability of DRH-1 protein. In the absence of virus, loss of *ulp-4* was sufficient to significantly diminish DRH-1 protein levels throughout the animal; concordantly, loss of *smo-1* resulted in

higher levels of DRH-1 (*mScarlet::DRH-1*; Fig. 3C). Red fluorescence was specific to the *mScarlet::DRH-1* transgene, as *drh-1(RNAi)* reduced fluorescence to almost undetectable levels (Fig. 3C). To confirm this finding, we conducted quantitative immunoblotting of DRH-1 after inactivation of either *ulp-4* or *smo-1*; basal levels of DRH-1 significantly decreased after *ulp-4(RNAi)* and increased after *smo-1(RNAi)* treatment (Fig. 3D,E). As expected, ULP-4 regulation of DRH-1 was strictly post-transcriptional, as the level of *drh-1* mRNA were unchanged after inactivation of *ulp-4* (Fig. 3F).

To gain insight into how ULP-4 regulated levels of DRH-1, we inactivated either the proteosome (*ubq-2(RNAi)*) or macroautophagy (*lgg-1(RNAi)*), hereafter referred to as autophagy), in conjunction with loss of *ulp-4* or *smo-1*, and assessed levels of DRH-1. *ubq-2* and *lgg-1* encode the orthologs of UBA52/ubiquitin and Atg8/LC3, which are required for proteosome function and autophagy, respectively (Kipreos, 2005; Melendez and Levine, 2009). Loss of *ubq-2* was sufficient to significantly increase

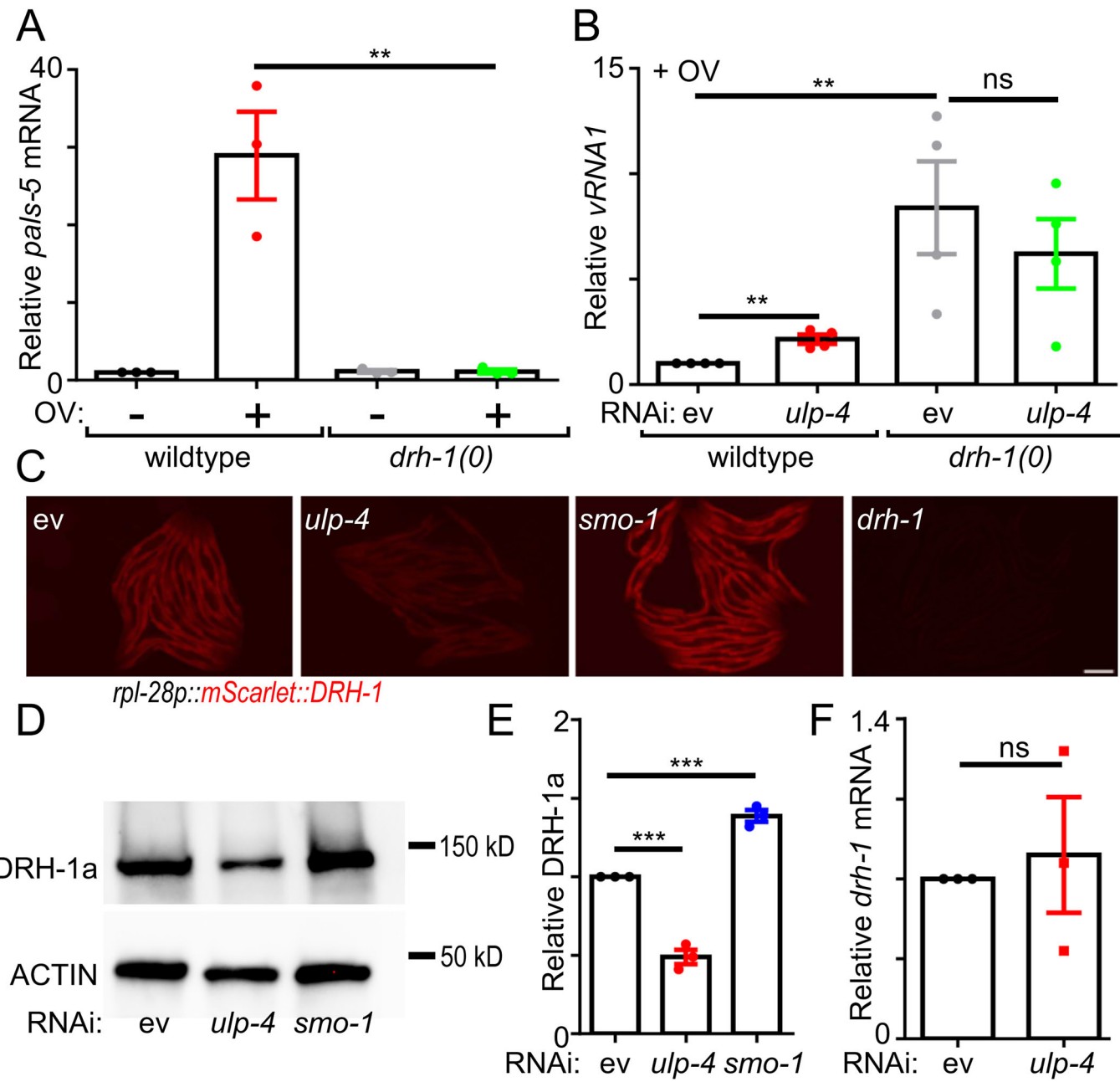

**Figure 3. ULP-4 promotes the protein stability of DRH-1 through deSUMOylation.**

(A) RT-qPCR analysis of endogenous *pals-5* in wildtype and *drh-1(ok3495)* null mutants +/− viral infection. **P = 0.0079. (B) RT-qPCR analysis of virus *RNA1* levels in wildtype and *drh-1(ok3495)* null animals treated with either empty vector or *ulp-4(RNAi)*, followed by viral infection. **P = 0.0019 (*ulp-4(RNAi)* in wildtype) P = 0.0152 (*drh-1(0)*), ns: P = 0.4568. (C) Representative images of *mScarlet::DRH-1* expression in uninfected animals after RNAi to empty vector, *ulp-4*, *smo-1*, or *drh-1*, respectively. Scale bar = 200 μm. (D) Representative immunoblot of conditions in panel (C). (E) Quantification of DRH-1 protein levels. ***P = 0.0004 (*ulp-4*) and P = 0.0005 (*smo-1*). (F) RT-qPCR analysis of endogenous *drh-1* after empty vector or *ulp-4(RNAi)*. NS: P = 0.6996. In all cases, values are the mean across three independent biological trials; error bars are SEM. A two-tailed *t*-test was used to calculate P values. Source data are available online for this figure.

fluorescence of the *mScarlet::DRH-1* transgene and DRH-1 protein levels in animals lacking *ulp-4, smo-1*, or that were otherwise wild-type (Fig. EV3A,B). While loss of either *smo-1* or *ubq-2* was sufficient to significantly increase DRH-1a levels, simultaneous inactivation of both had no additive effect (Fig. EV3C), which suggests that SUMOylation promotes DRH-1 proteosomal

degradation. However, in the absence of *ulp-4*, inactivation of the proteosome did not fully elevate DRH-1 levels (Fig. EV3C). We next tested whether SUMOylation regulated DRH-1 through autophagy; loss of autophagy did not significantly alter levels of DRH-1 (Fig. EV3). Next, we tested whether infection was sufficient to stabilize DRH-1. We found that DRH-1 levels were unchanged

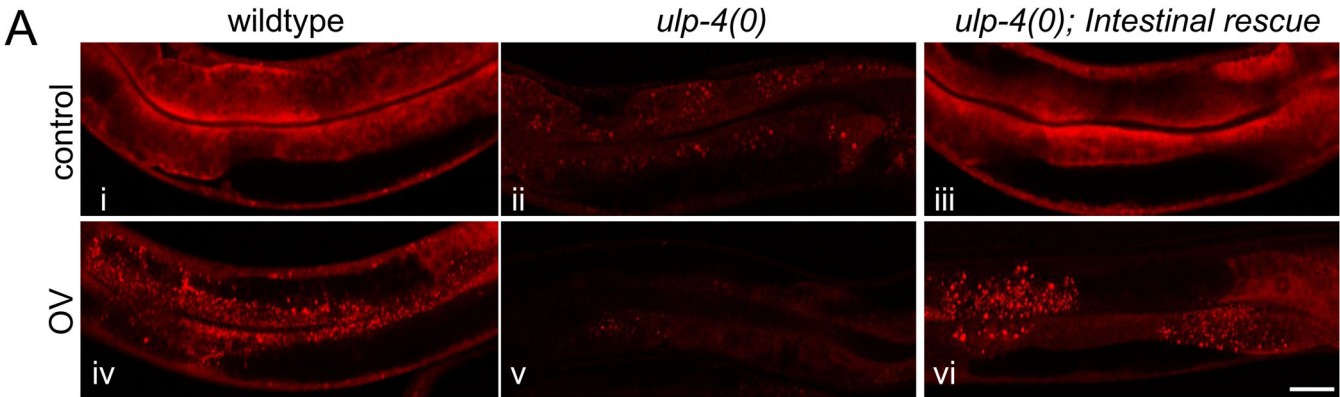

**Figure 4.  NonSUMOylated DRH-1 translocates to the mitochondria upon viral infection.**

(A) Representative images of *mScarlet::DRH-1* in the intestinal region of wildtype, *ulp-4(tm3688)*, or *ulp-4(tm3688);artEx95(ges-1p::ULP-4)* of uninfected (i–iii) or virally infected animals (iv–vi). Scale bar = 20 µm. (B) Representative images of an intestinal cell of a *mScarlet::DRH-1;mito-GFP* animal infected with virus; lysosomal-related organelles/gut granules are indicated (405-nm blue channel). Scale bar = 2 µm. (C) Representative image of an intestinal cell of a *mScarlet::DRH-1;GFP::SMO-1* animal infected with virus; mitochondria are indicated (Mitotracker, Far Red). Arrows highlight DRH-1 localization to the mitochondria periphery. Scale bar = 2 µm. Source data are available online for this figure.

by viral infection (Appendix Fig. S2); however, OV typically only infects a few intestinal cells (Franz et al, 2014), and OV may only stabilize DRH-1 within infected cells (assessed below). Under basal conditions, DRH-1 co-localized with SMO-1 throughout the cytosol, which suggests SMO-1 conjugation to DRH-1 may not be sufficient for degradation (Appendix Fig. S3). Collectively, we conclude DRH-1 is degraded via the proteosome, which is influenced in part through SUMOylation. However, ULP-4 acts upstream of the proteosome to stabilize DRH-1, either directly or indirectly, through an additional post-transcriptional mechanism.

## NonSUMOylated DRH-1 translocates to mitochondria upon viral infection

To further elucidate how SUMOylation regulates DRH-1, we assessed the localization of DRH-1 following viral infection. Consistent with the findings of others (Batachari et al, 2024), viral infection initiated a shift in DRH-1 from a diffuse cytosolic to the formation of puncta (Fig. 4A). In the absence of *ulp-4*, DRH-1 levels (mScarlet::DRH-1) were almost undetectable, but restoring *ulp-4* expression within the intestine was sufficient to restore DRH-1 induction and the puncta formation following viral infection (Fig. 4A), which suggests ULP-4 acts cell autonomously, perhaps directly upon DRH-1, to regulate the IPR. To determine the subcellular localization of DRH-1 containing puncta, we co-expressed *mScarlet::DRH-1* with GFP fused to a mitochondrial localization tag (*ges-1p::GFP::MITO*). Following viral infection, we observed that DRH-1 primarily co-localized with mitochondria within infected intestinal cells (Fig. 4B), To a much lesser extent, some overlap with auto-fluorescent gut granules were also observed (Fig. 4B), which have previously been shown to be lysosome-related organelles (LROs) (Coburn and Gems, 2013). Next, we determined whether the SUMOylation status of DRH-1 dictated subcellular localization. Using transgenic animals co-expressing *GFP::SMO-1*, *mScarlet::DRH-1*, and treated with Mitotracker, we found little to no overlap between the SUMO moiety (SMO-1) and DRH-1 at puncta that form during viral infection (Fig. 4C). Collectively, our findings indicate that unSUMOylated DRH-1 translocates to mitochondria following viral infection, and this is mediated by the deSUMOylating activity of ULP-4.

## deSUMOylation of DRH-1 activates antiviral defense

We posit ULP-4 may directly deSUMOylate DRH-1 to activate the IPR. SUMOylation frequently occurs at a consensus sequence (ψKxE). We used four different prediction tools to identify potential SUMOylation sites in DRH-1 (Beauclair et al, 2015; Marongiu et al, 2010; Wang et al, 2020; Zhao et al, 2014). Among the 60 lysine residues in DRH-1, all four algorithms only predicted K647 and K731 as targets for SUMOylation with high confidence

(Appendix Fig. S4A; Dataset EV1). While K647 does not fall within a specific domain, K731 is located within the helicase domain. All RIG-I-like receptors (RLR) have a central helicase domain and a carboxy-terminal domain (CTD), which work together to detect immunostimulatory RNAs (Rehwinkel and Gack, 2020). To functionally test whether K647/731 are the SUMOylation sites of DRH-1, both were mutated to arginine, which prevents SUMOylation at those residues. Transgenic animals were generated that express either the mutated (2KR) or wildtype isoform of DRH-1a throughout the soma (*rpl-28p::mScarlet::DRH-1(2KR)*); transgenes were expressed in *drh-1(ok3495)* null mutants at similar levels, approximately fourfold higher than endogenous *drh-1* (Appendix Fig. S4B). Protein levels of DRH-1 and DRH-1(2KR) were also similar in vivo (Fig. 5A–C). Consistent with the findings of others (Batachari et al, 2024), overexpression of wild-type *drh-1* was sufficient to modestly activate the IPR in vivo in the absence of virus (*pals-5p::GFP*; Fig. 5A). In contrast, DRH-1(2KR) animals robustly induced *pals-5p::GFP* fluorescence (Fig. 5B), despite having slightly lower levels of red fluorescence (mScarlet::DRH-1(2KR); Fig. 5C). Loss of *ulp-4* suppressed the induction of the IPR in animals overexpressing the wild-type isoform of DRH-1 (Fig. 5A). In contrast, loss of *ulp-4* had no effect on the induction in animals expressing *drh-1(2KR)* (Fig. 5B). Similar results were found with expression of endogenous *pals-5* mRNA (Fig. 5D), and of another gene induced by viral infection (*sdz-6*; Appendix Fig. S4C) (Chen et al, 2017). Next, we identified the subcellular distribution of DRH-1 and DRH-1(2KR) under basal conditions. As expected, in the absence of viral infection, wild-type DRH-1 was predominantly diffuse throughout the cytosol, but overexpression led to the formation of some puncta (Fig. 5E; Appendix Fig. S5), which is consistent with modest induction of the IPR. In contrast, mScarlet::DRH-1(2KR) was predominantly localized in puncta either surrounding mitochondria or at LROs neighboring mito-chondria (Figs. 5F,G and EV4; Appendix Fig. S5); the latter observation suggests DRH-1 activation may have antiviral activity through the coupling mitochondrial and lysosomal activity. Collectively, we conclude that SUMOylation normally limits activation of DRH-1 and that upon viral infection, ULP-4 deSUMOylates DRH-1, allowing translocation to the mitochondria to activate innate antiviral defenses.

## Animals with compromised antiviral defense fail to clear the virus and show signs of pathogenesis

To test the physiological consequences of *ulp-4* or *drh-1* loss, we assessed intestinal status using a tagged GFP that localizes at the apical membrane of the intestine to mark the lumen. In the absence of viral infection, loss of either *ulp-4* or *drh-1* had no effect on the lumen (Fig. 6Ai–iii). Similarly, viral infection was not sufficient to cause obvious changes to the intestinal lumen (Fig. 6Aiv). However,

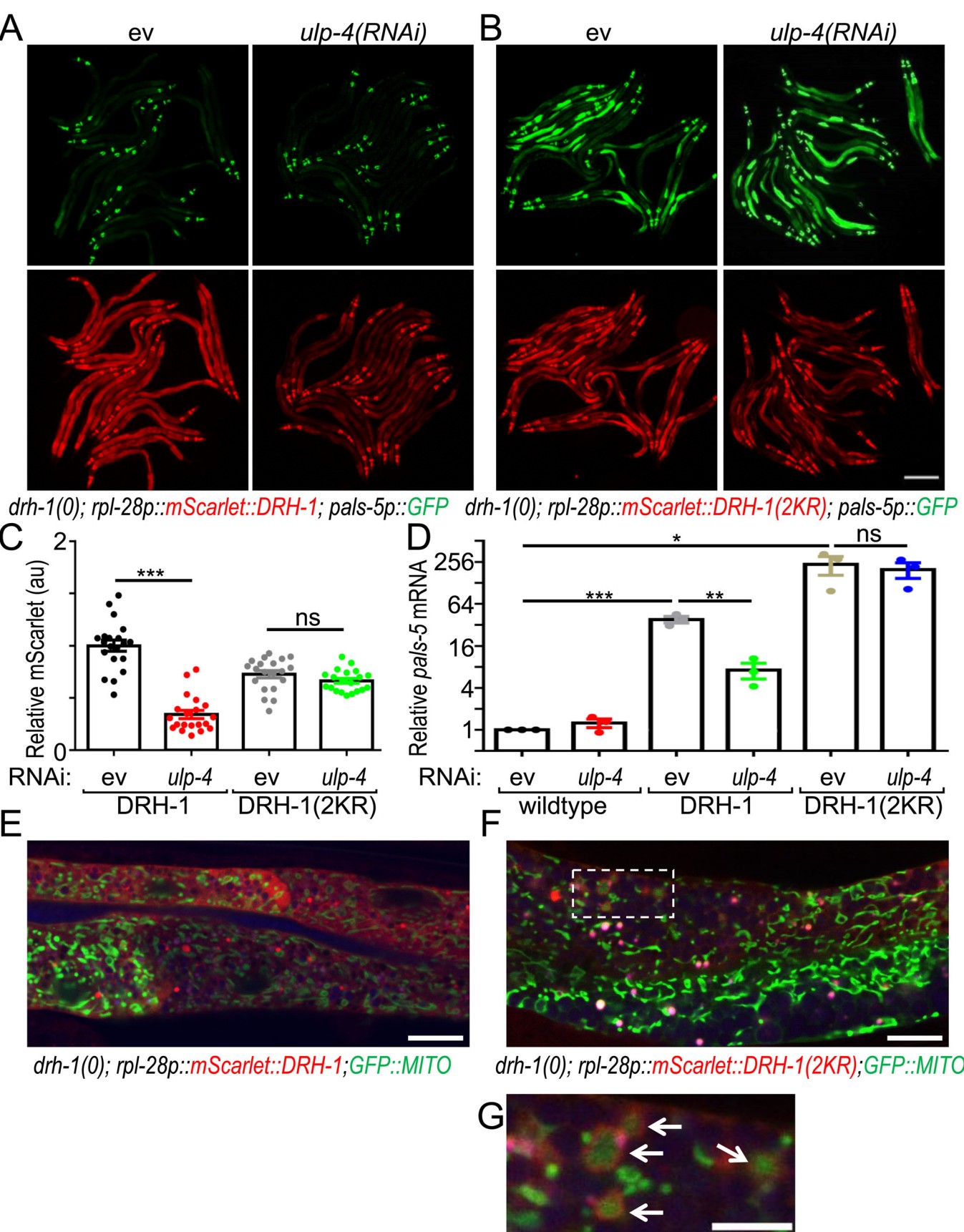

drh-1(0); rpl-28p::mScarlet::DRH-1; pals-5p::GFP

drh-1(0); rpl-28p::mScarlet::DRH-1(2KR); pals-5p::GFP

drh-1(0); rpl-28p::mScarlet::DRH-1;GFP::MITO

drh-1(0); rpl-28p::mScarlet::DRH-1(2KR);GFP::MITO

**Figure 5. deSUMOylation of DRH-1 activates antiviral defense.**

(A, B) Representative images of *pals-5p::GFP* and *mScarlet* expression in *drh-1(ok3495)* plus somatic overexpression of either wild-type DRH-1 (*rpl-28p::mScarlet::DRH-1(wt)*) (A), or non-SUMOylatable DRH-1 (*rpl-28p::mScarlet::DRH-1(2KR)* (B), treated with either empty vector or *ulp-4(RNAi)*. Scale bar = 200 µm. (C) Quantification of mScarlet::DRH-1 in (A, B). \*\*\*P < 0.0001, NS: P = 0.1412. (D) RT-qPCR analysis of endogenous *pals-5* of samples from (A, B) and *jyls8* wild-type control. \*P = 0.0293, \*\*P = 0.0024, \*\*\*P = 0.0009, ns: P = 0.6911. (E–G) Representative composite images of uninfected L4 animals expressing *mScarlet::DRH-1(WT);GFP::mito* (E), or *mScarlet::DRH-1(2KR);GFP::mito* (F, G), respectively. Arrows highlight DRH-1 localization to the mitochondria periphery. Scale bar = 2 µm. Inset scale bar = 5 µm. Images are reused in Appendix Fig. S5, with individual fluorescence panels. In all cases, values are the relative mean from three independent biological trials; error bars are the SEM. A two-tailed *t*-test was used to calculate *P* values. Source data are available online for this figure.

in the absence of either *ulp-4* or *drh-1*, viral infection produced signs of intestinal bloating in a significant portion of infected animals (Fig. 6Av, vi,B). Intestinal bloating has previously been shown to occur during infection with bacterial pathogens and is indicative of increased pathogenesis (Singh and Aballay, 2019). Next, we tested whether loss of DRH-1 impacted viral load over time, a surrogate measure of viral clearance. Wild-type animals had a significant reduction of viral levels four days post-infection. In contrast, animals lacking either *ulp-4* or *drh-1* not only have higher viral loads than wildtype, but *ulp-4* and *drh-1* null mutant animals both failed to significantly decrease viral loads over time (Fig. 6C). Thus, ULP-4 and DRH-1 play a critical role in activating the IPR in response to viral infection and that loss of this response results in increased pathogenesis and an inability to clear the virus.

## Antiviral response declines during aging in part due to dysregulation of DRH-1 SUMOylation

Aging results in the diminished inducibility of a myriad of adaptive transcriptional responses that maintain homeostasis following stress (Haigis and Yankner, 2010; Pomatto and Davies, 2017; Vilchez et al, 2014). We tested whether the inducibility of the IPR declined during normal aging and the role of SUMOylation. By the 4th day of adulthood, the induction of the IPR by viral infection was severely impaired in wild-type animals (*pals-5p::GFP*, *pals-5* mRNA; Fig. 7A,B). Because levels of viral load tend to scale with levels of *pals-5::GFP* induction, we determined whether older animals failed to induce the IPR due to lower viral loads. Young and older animals had similar viral loads (Fig. EV5A); thus, older animals were unable to mount a full antiviral response. At this age, transcriptional responses to diverse forms of stress are known to be impaired (Ben-Zvi et al, 2009; Denzel et al, 2019; Dues et al, 2016; Labbadia and Morimoto, 2015a, b) and global patterns of hyper-SUMOylation within the proteome have been reported (Princz et al, 2020). We found that the inducibility of *pals-5* in response to heat, a non-viral trigger, was also lost (Fig. EV5B). However, levels of DRH-1 protein are decreased by the 4th day of adulthood (Fig. 7C,D), which suggests DRH-1 regulatory mechanisms may also decline with aging. Constitutive *drh-1* expression throughout the intestine of *drh-1* null animals failed to offset the age-associated decline in the IPR, despite fully rescuing IPR induction after viral infection in young animals (Fig. 7E).

To test whether the proper regulation of DRH-1, but not levels of DRH-1 per se, breakdown during aging, we first examined whether levels of *ulp-4* were altered in older animals. Expression of *ulp-4* mRNA significantly declined by day 4 of adulthood (Fig. 7F). Thus, an age-associated decline of *ulp-4* may compromise the IPR by failing to remove SMO-1 from DRH-1. First, we tested this by

intestinal overexpression of *drh-1(2KR)*. In contrast to wild-type DRH-1 overexpression, older animals expressing intestinal DRH-1(2KR) induced the IPR approximately 50-fold after infection (Fig. 7E), suggesting partial rescue. It is worth noting that while infecting young *drh-1(2kr)* animals resulted in a 2.4-fold hyper-induction of the IPR (Fig. 7E, column 6/column 4), the relative induction in older animals was significantly greater (sixfold, p = 0.02 Fig. 7E, column 12/column 10), which is consistent with rescue of age-associated decline. Unlike intestinal rescue, expression of *drh-1* or *drh-1(2kr)* throughout the soma was sufficient to induce the IPR in the absence of virus in young animals (Fig. 7G,H, respectively). During aging, uninfected DRH-1(2KR) animals had increasing expression of *pals-5* (Fig. 7E,H,I). In contrast, levels of *pals-5* decreased during aging in animals overexpressing wild-type DRH-1 throughout the soma (Fig. 7G), despite wild-type DRH-1 being expressed at higher levels than DRH-1(2KR) (Fig. 7J). Collectively, we conclude that aged *C. elegans* are unable to induce the IPR, at least in part, due to dysregulation of DRH-1 through increased SUMO conjugation; preventing this dysregulation revealed that *aging itself* induces an antiviral response.

## Discussion

### ULP-4 deSUMOylates DRH-1 to prevent proteosomal degradation

In mammals, the deconjugating activity of SENP family members are specific: SENP1&2 remove all SUMO moieties (i.e., SUMO1-5) (Claessens and Vertegaal, 2024). In contrast, SENP6&7 are only responsible for removing SUMO2/3 (Claessens and Vertegaal, 2024; Kolli et al, 2010; Kunz et al, 2018; Mikolajczyk et al, 2007). SUMO2/3 form covalent polymers that are recognized by SUMO-targeted ubiquitin ligases (STUbLs), which conjugate ubiquitin to form hybrid chains (Wang and Matunis, 2023). SUMO-Ub hybrid chains target substrates for proteasomal degradation, but can also mediate nonproteolytic signaling (Fryrear et al, 2012; Jansen and Vertegaal, 2021; Poulsen et al, 2013). In contrast, SUMO1 cannot form monomeric chains (Tatham et al, 2001). These structural differences are thought to be a major driver of the functional differences between SUMO1 and SUMO2/3 (Ma et al, 2023; Wang and Matunis, 2023). *C. elegans* SMO-1 has similarities to both SUMO1 and SUMO2/3: the sequence is more similar to SUMO1, but the electrostatic surface features are closer to SUMO2/3 (Surana et al, 2017). SMO-1 conjugation is thought to execute all of the functions of mammalian SUMO1 and SUMO2/3, including the formation of novel non-covalent protein–protein interactions through typical SUMO Interacting Motifs (SIMs) (Surana et al,

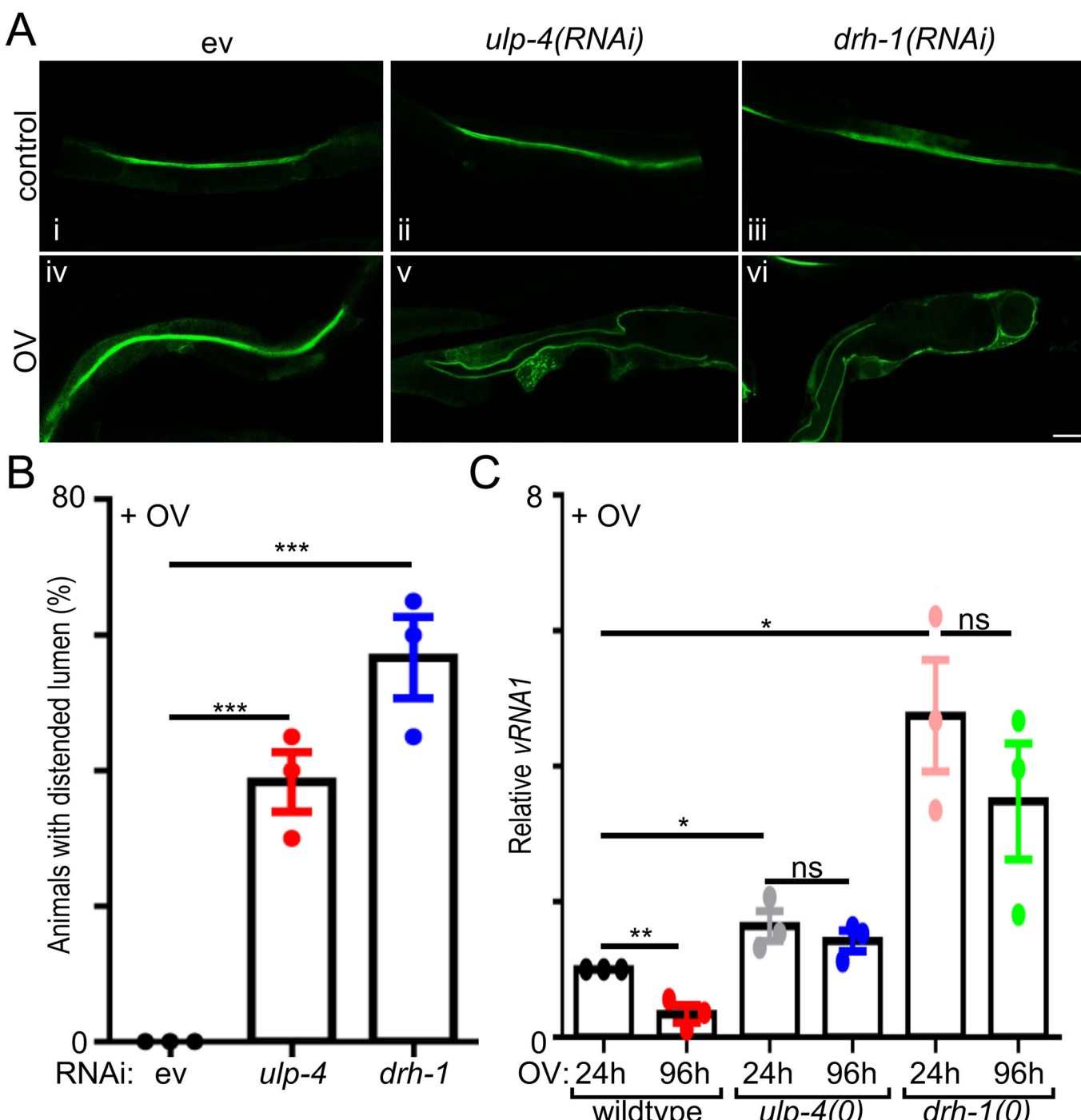

**Figure 6. Antiviral defense through ULP-4 and DRH-1 limits viral pathogenesis.**

(A) Representative images of animals expressing the *ACT-5::GFP* apical membrane marker treated with either empty vector, *ulp-4(RNAi)*, or *drh-1(RNAi)*, respectively, +/− viral infection. Scale bar = 20 μm. (B) Quantification of the portion of a population with an enlarged lumen after viral infection. ***$P = 0.0010$ (*ulp-4*) and $P = 0.0007$ (*drh-1*). (C) RT-qPCR quantification of Orsay virus *RNA1* in wildtype, *ulp-4(tm3688)*, or *drh-1(ok3495)* null mutant animals infected with the virus. *$P = 0.0446$ (*drh-1(0)*, 24 h), $P = 0.0105$ (*ulp-4(0)*, 24 h), **$P = 0.0068$, ns: $P = 0.3480$ (*ulp-4(0)*, 96 h) and $P = 0.4618$ (*drh-1(0)*, 96 h). In all cases, values are the mean of three independent biological trials; error bars are the SEM. A two-tailed *t*-test was used to calculate *P* values. Source data are available online for this figure.

2017). SMO-1 conjugation has been shown to alter protein–protein interactions, initiate shifts in subcellular localization, and regulate chromatin and transcription factors (Broday, 2017; Drabikowski, 2020; Pelisch et al, 2014). For instance, we previously found that

SUMOylation of the heat shock transcription factor (HSF-1) limits the activation of the heat shock response in *C. elegans* (Das et al, 2017).

Our genetic analysis supports the notion that ULP-4, orthologous to SENP7, prevents the SMO-1-mediated proteosomal

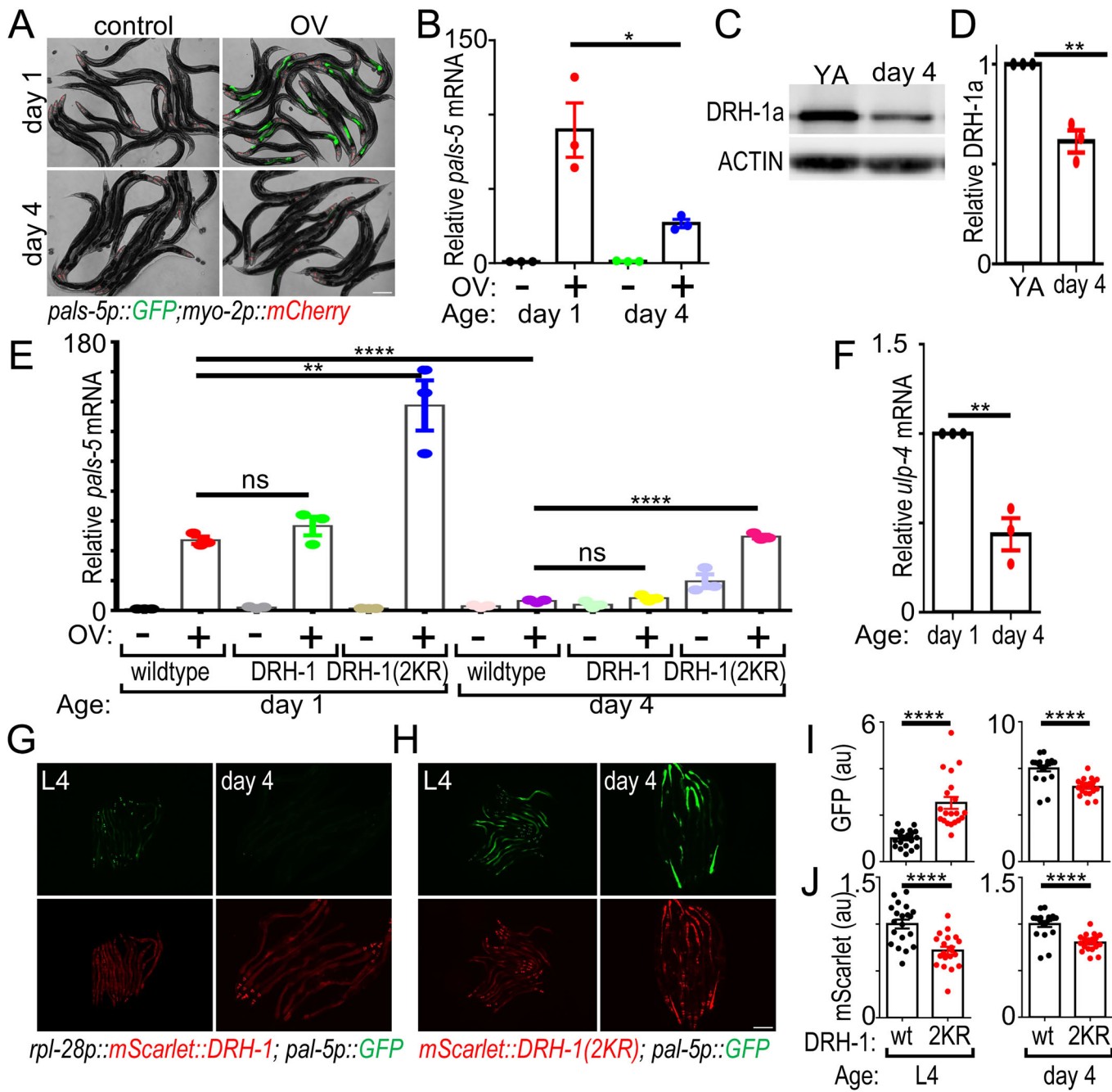

**Figure 7. The IPR declines during aging due to dysregulation of DRH-1 SUMOylation.**

(A) Representative images of *pals-5p::GFP* expression $+/-$ viral infection. Viral infection was initiated at L4 or day 3 of adulthood, and images were obtained 24 h later. Scale bar = 200 μm. (B) Quantification of GFP fluorescence intensity of (A). *$P = 0.0276$. (C) Representative immunoblot of mScarlet::DRH-1 levels in young adult (YA) or day 4 adult animals. (D) Quantification of DRH-1 protein levels. **$P = 0.0022$. (E) RT-qPCR analysis of endogenous *pals-5* $+/-$ viral infection at different ages in wildtype, or *drh-1(ok3495)* null mutant animals with restored intestinal expression (*ges-1p*) of either wild-type DRH-1 (*mScarlet::DRH-1(wt)*) or non-SUMOylatable DRH-1 (*mScarlet::DRH-1(2KR)*). **$P = 0.0059$, ****$P < 0.0001$, ns: $P = 0.2286$ (day 1) and $P = 0.2122$ (day 4). (F) RT-qPCR analysis of endogenous *ulp-4* at different ages in wildtype animals. **$P = 0.0034$. (G, H). Representative images of *pals-5p::GFP* expression (top) and *mScarlet* (bottom) at different ages in *drh-1(ok3495)* null mutant animals with restored somatic expression (*rpl-28p*) of either wild-type DRH-1 (*mScarlet::DRH-1(wt)*) or non-SUMOylatable DRH-1 (*mScarlet::DRH-1(2KR)*). (I) Quantification of relative *pals-5::GFP* fluorescence in panels (G, H). ****$P < 0.0001$. (J) Quantification of relative *mScarlet* fluorescence in panels (G, H). ****$P < 0.0001$. In all cases, values are the relative mean of 20 animals per trial, across three independent biological trials; error bars are the SEM. A two-tailed *t*-test was used to calculate *P* values. Source data are available online for this figure.

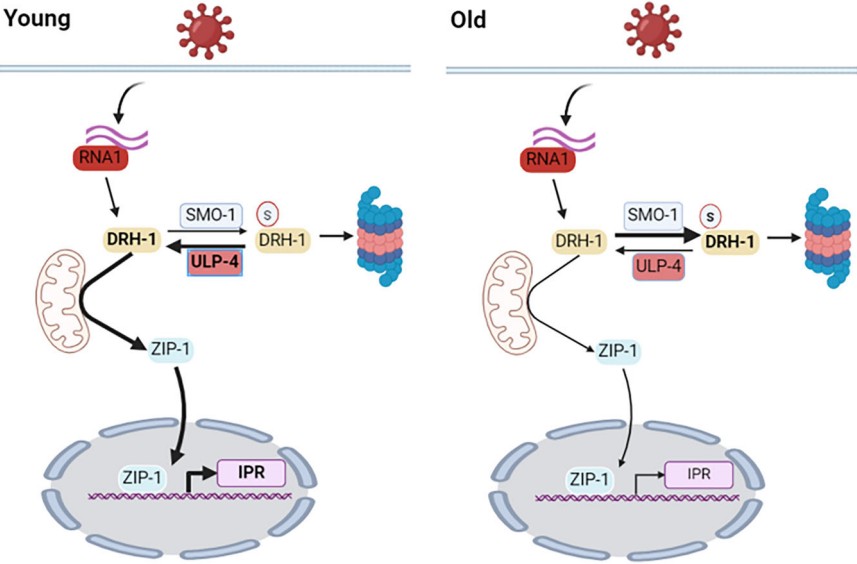

**Figure 8. Model of dysregulation of SUMO conjugation to DRH-1 during aging.**

Age-associated loss of *ulp-4* compromises antiviral innate immunity to promote pathogenesis, which occurs at least in part through increased DRH-1 degradation via the proteasome.

degradation of DRH-1, which acts as a regulatory mechanism that limits antiviral innate immunity (Fig. 8). However, additional mechanisms are liable to regulate DRH-1. We were unable to directly observe SUMOylated DRH-1, which is likely due to technical limitations: SUMO is a transient modification, only a few infected intestinal cells would be affected, and SMO-1 conjugated DRH-1 may be quickly degraded under the right conditions. Inhibition of the proteosome did not fully restore DRH-1 levels in the absence of *ulp-4*, which suggests additional regulatory components influence DRH-1 levels, localization, and activity. Consistent with this notion, mammalian RIG-I is regulated through additional post-translational modifications including ubiquitination, phosphorylation, ISGylation, and acetylation (Chiang and Gack, 2017; Loo and Gale, 2011; Oshiumi, 2020; Villarroya-Beltri et al, 2017; Wang et al, 2023).

In mammals, the role of SUMOylation in viral innate immunity through activation of the type I IFN response is complicated. RIG-I has been shown to be SUMOylated by SUMO1 in response to RNA viral infection, which stabilizes RIG-I to enhance IFN-I production by increasing interaction with Mitochondrial Antiviral-Signaling (MAVS) proteins (Hu et al, 2017; Mi et al, 2010). In stark contrast, SUMO2 and SUMO3 have previously been shown to act redundantly through an unknown mechanism to potently inhibit the activation of the Type I IFN response (Crowl and Stetson, 2018). Our results hint that the Type I IFN response may be negatively or positively regulated by SUMO2/3 or SUMO1, respectively, by converging upon RIG-I. Alternatively, the opposing roles of SUMO1 and SUMO2/3 in regulating antiviral innate immunity may have diverged; mammalian SENP isopeptidases also have opposing roles in viral innate immunity. SENP1 deconjugates SUMO1 to negatively regulate antiviral immunity by deSUMOylation of MAVS and activation of IRF3 (Dai et al, 2023). In contrast, SENP7

deSUMOylates cGAS, which induces expression of IRF3-responsive genes (Cui et al, 2017).

## Mitochondria are evolutionarily conserved antiviral-signaling hubs

Mitochondria act as subcellular signaling hubs in response to stress, including viral infections (Shen et al, 2022). The type of response is dictated both by the nature and threshold of the stress; for example, low of levels of stress within the mitochondrial proteome activate the mitochondrial unfolded protein response (UPR$^{mito}$), a molecular mechanism to restore homeostasis that preserves the cell. Increasing levels of stress activate subcellular responses, including mitochondrial fusion to dilute damage and fission to remove damage that gets sequestered at the ends of filament networks, which can progress to mitophagy (Baker et al, 2011; Shen et al, 2022). In mammals, chronic or high levels of stress induce programmed cell death pathways to preserve tissue homeostasis; the most well-studied is apoptosis, which is triggered from mitochondria (Lee et al, 2023). In mammals, mitochondria also function as a platform for the innate immune response to infection by RNA viruses. Innate immunity in mammals relies on MAVS proteins on the outer mitochondrial membrane. Viral RNA binds to RIG-I and related RLRs to activate viral innate immune signaling (Kell and Gale, 2015; Onomoto et al, 2021; Rehwinkel and Gack, 2020; Thoresen et al, 2021). Despite the lack of a MAVS ortholog in *C. elegans*, an analogous response is occurring, as DRH-1 translocates to the mitochondria after viral infection, and mitochondria still serve as the focal point for initiating antiviral response. Our results open the possibility that RIG-I or other mammalian RIG-I paralogs may localize to mitochondria independent of MAVs interactions, but reinforce the notion that mitochondria play a central role in antiviral innate immunity,

even though evolutionary pressure between viruses and hosts may have favored a divergence in the detailed regulation of antiviral components.

A role linking SENP7 and mitochondrial function in mammals has not been described, but emerging results in *C. elegans* support a possible link. The handful of *C. elegans* studies on ULP-4 have implicated a key role for coordinating stress response initiated in mitochondria (Gao et al, 2019; Sapir et al, 2014). First, ULP-4 has been shown to regulate the first enzyme of the mevalonate pathway, which undergoes age-dependent SUMOylation and correlates with a shift in ULP-4 subcellular localization from the cytosol to the mitochondria in non-intestinal tissues (Sapir et al, 2014). In another study, ULP-4 was shown to be a positive regulator of DVE-1 and ATFS-1, two transcription factors that are the primary effectors of the UPR$^{mito}$ (Gao et al, 2019). In the absence of *ulp-4*, DVE-1 failed to translocate to the nucleus and the induction of the UPR$^{mito}$ is impaired (Gao et al, 2019). Viral targeting of mitochondria generates stress that induces retrograde signals (Foo et al, 2022; Sorouri et al, 2022), and our findings that ULP-4 deSUMOylates DRH-1 to facilitate mitochondrial localization and activation of the IPR suggest this may be conserved in *C. elegans*. We find it intriguing that ULP-4 regulates both the UPR$^{mito}$ and IPR, two mitochondrial signal transduction pathways that maintain homeostasis. How target selectivity of SENPs is obtained remains a mystery, but given the relatively small number of SUMO isopeptidase genes, it has been postulated that groups of proteins functionally and physically linked are regulated by the same SENP (Nayak & Müller, 2014). For example, SUMO often targets multiple proteins within a complex, or within a pathway (Chymkowitch et al, 2015; Hendriks et al, 2014). It will be important in future studies to identify how ULP/SENP target selectivity is obtained to maintain homeostasis in animals experiencing diverse forms of stress linked to mitochondria.

### Dysregulated SUMO conjugation of DRH-1 during aging compromises antiviral innate immunity to promote pathogenesis

*C. elegans* provides an opportunity to determine how antiviral responses in non-immune cells change with age, which would be difficult to discover in vivo in more complex metazoans. While incredibly complex, immunosenescence in mammals is well-described and can be most simply defined as the gradual deterioration of both the acquired and innate immune systems during aging. Immunosenescence is also associated with the onset of a chronic, systemic inflammatory state (i.e., inflammaging) (Feehan et al, 2021; Lopez-Otin et al, 2013, 2023; Nikolich-Zugich, 2018). The study of chronic viral infections in the elderly that are attributed to declining innate immunity has largely focused on neutrophils, monocytes, macrophages, natural killer cells, natural killer T (NKT) cells, and dendritic cells (Gomez et al, 2008; Montgomery and Shaw, 2015; Panda et al, 2009; Stotesbury and Sigal, 2021). Our findings that loss of the IPR results in a chronic viral infection and produces signs of pathogenesis in *C. elegans* is consistent with the possibility that the aging of non-immune cells in higher vertebrates is an additional level of immunosenescence.

Our discovery that the inducibility of the innate immune response declines due to the loss of ULP-4-mediated deSUMOylation of DRH-1 represents a novel age-associated breakdown in

antiviral defense, which is likely distinct from the well-characterized age-associated decline in transcriptional stress responses associated with changes in chromatin. Our discovery that rescuing DRH-1 activity in aged animals was sufficient to induce the IPR in the absence of virus is consistent with findings that downregulation of retrotransposons delays *C. elegans* aging and emerging evidence that activation of latent viral elements promotes aging in complex metazoans, including humans (Gorbunova et al, 2021; Sturm et al, 2023). In *C. elegans*, several published studies, including two from our laboratory, have implicated either SMO-1 or orthologs of SENPs in aging or the regulation of pathways associated with healthy aging (Das et al, 2017; Gao et al, 2019; Princz et al, 2020; Samuelson et al, 2007; Sapir et al, 2014), but viral innate immunity had not been previously investigated. Emerging evidence suggests that an age-associated breakdown in proper SUMOylation of the proteome contributes to aging and age-associated disease in humans. SUMOylation plays a key role in protein quality control and the maintenance of proteostasis and has been implicated in nearly all age-associated neurological disorders (Wang and Matunis, 2023). Our discoveries suggest a possible link between the age-associated loss of antiviral defense and a shift in the physiological state of non-immune cells, which may contribute to the accumulation of endogenous retroviral elements and increased viral pathogenesis in the elderly.

## Methods

**Reagents and tools table**

| Experimental models | | |
| --- | --- | --- |
| **E. coli and virus** | **Source** | **Identifier** |
| HT115 (*E. coli*) | Caenorhabditis Genetics Center | N/A |
| OP50 (*E. coli*) | Caenorhabditis Genetics Center | N/A |
| Orsay virus | Troemel laboratory | N/A |
| **C. elegans strains** | **Source** | **Identifier** |
| N2 Bristol | Caenorhabditis Genetics Center | N/A |
| *jyIs8[pals-5p::GFP+myo-2p::mCherry]* | Caenorhabditis Genetics Center | ERT54 |
| *ulp-4(tm3688)/ mIn1[mIs14dpy-10(e128)]; jyIs8[pals-5p::GFP + myo-2p::mCherry]* | This study | AVS876 |
| *ulp-4(tm3688)/ mIn1[mIs14dpy-10(e128)]; jyIs8[pals-5p::GFP+myo-2p::mCherry]; artEx95[ges-1p::ulp-4+myo-2p::mCherry]* | This study | AVS877 |
| *ulp-4(tm3688)/mIn1[mIs14 dpy-10(e128)] ; jyIs8[pals-5p::GFP+myo-2p::mCherry]; artEx96[myo-3p::ulp-4+ myo-2p::mCherry]* | This study | AVS878 |
| *ulp-4(tm3688)/mIn1[mIs14 dpy-10(e128)] ; jyIs8[pals-5p::GFP+myo-2p::mCherry]; artEx104[dpy-7p::ulp-4+ myo-2p::mCherry]* | This study | AVS912 |
| *ulp-4(tm3688)/mIn1[mIs14 dpy-10(e128)]; jyIs8[pals-5p::GFP+myo-2p::mCherry]; artEx103[rab-3p::ulp-4+ myo-2p::mCherry]* | This study | AVS911 |

| Experimental models | | |
|---|---|---|
| **E. coli and virus** | **Source** | **Identifier** |
| rde-1(ne219) V; kbIs7 [nhx-2p::rde-1 + rol-6(su1006)] | Caenorhabditis Genetics Center | VP303 |
| gei-17(tm2723)/hT2 [bli-4(e937) let-?(q782) qIs48] I; jyIs8 [pals-5p::GFP + myo-2p::mCherry] | This study | AVS881 |
| drh-1(ok3495); jyIs8[pals-5p::GFP+ myo-2p::mCherry] | This study | AVS882 |
| mgTi54[rpl-28p::mScarlet::drh-1] | Ruvkun laboratory | GR3325 |
| artIs14[rpl-28p::mScarlet::drh-1+myo-2p::gfp] | This study | AVS928 |
| ulp-4(tm3688)/mIn1 [mIs14 dpy-10(e128)]; artIs14[rpl-28p::mScarlet::drh-1+myo-2p::gfp] | This study | AVS929 |
| ulp-4(tm3688)/mIn1 [mIs14 dpy-10(e128)]; artIs14[rpl-28p::mScarlet::drh-1+myo-2p::gfp]; artEx95[ges-1p::ulp-4 + myo-2p::mCherry] | This study | AVS932 |
| artIs14[rpl-28p::mScarlet::drh-1+myo-2p::gfp]; zcIs17[ges-1p::GFP(mit)]. | This study | AVS1036 |
| artIs14[rpl-28p::mScarlet::drh-1+myo-2p::gfp]; [smo-1p::8XHIS::GFP::smo-1] | This study | AVS1037 |
| drh-1(ok3495); jyIs8[pals-5p::GFP+myo-2p::mCherry]; artEx106[rpl-28p::mScarlet::drh-1+myo-2p::gfp] | This study | AVS1038 |
| drh-1(ok3495); jyIs8[pals-5p::GFP+ myo-2p::mCherry]; artEx107[rpl-28p::mScarlet::drh-1(k647R + K731R)+ myo-2p::gfp] | This study | AVS1039 |
| artEx106[rpl-28p::mScarlet::drh-1+myo-2p::gfp]; zcIs17 [ges-1p::GFP(mit)] | This study | AVS1040 |
| artEx107[rpl-28p::mScarlet::drh-1(k647R + K731R)+myo-2p::gfp]; zcIs17 [ges-1p::GFP(mit)] | This study | AVS1041 |
| jyIs13[act-5p::GFP::ACT-5+rol-6(su1006)] | Caenorhabditis Genetics Center | ERT60 |
| drh-1(ok3495); jyIs8[pals-5p::GFP+myo-2p::mCherry]; artIs11[ges-1p::6XHIS::drh-1+myo-2p::gfp] | This study | AVS914 |
| drh-1(ok3495); jyIs8[pals-5p::GFP+ myo-2p::mCherry]; artIs12[ges-1p::6XHIS::drh-1(k647R+K731R)+myo-2p::gfp] | This study | AVS926 |
| **Recombinant DNA** | **Source** | |
| ges-1p::ulp-4 | Our laboratory | |
| rab-3p::ulp-4 | Our laboratory | |
| dpy-7p::ulp-4 | Our laboratory | |
| myo-2p::ulp-4 | Our laboratory | |
| rpl-28p::mScarlet::drh-1 | Ruvkun laboratory | |

| Experimental models | | |
|---|---|---|
| **E. coli and virus** | **Source** | **Identifier** |
| rpl-28p::mScarlet::drh-1(k647R + K731R) | Our laboratory | |
| ges-1p::6XHIS::drh-1 | Our laboratory | |
| ges-1p::6XHIS::drh-1(k647R + K731R) | Our laboratory | |
| **Antibodies** | **Source** | **Catalog number** |
| anti-RFP | Rockland | # 600-401-379 |
| anti-actin | CST | #4967 |
| **Oligonucleotides** | | |
| **Name** | **Sequence** | |
| cdc-42-F | CGACAATTACGCCGTCACAG | |
| cdc-42-R | AAACACGTCGGTCTGTGGAT | |
| snb-1-F | CCGGATAAGACCATCTTGACG | |
| snb-1-R | GACGACTTCATCAACCTGAGC | |
| pals-5-F | GGACTATGTGAGCATATTGTTTCC | |
| pals-5-R | AGCGTTTTTCTGCACTGGTT | |
| RNA1-F | ACCTCACAACTGCCATCTACA | |
| RNA1-R | GACGCTTCCAAGATTGGTATTGGT | |
| drh-1-F | GCGATTTTCACTACATTCTCGA | |
| drh-1-R | TTCTTCGTTGCGCTCTATTTC | |
| ulp-4-F | CGGAACTCATCTAGATGGTTCTATC | |
| ulp-4-R | CCGCTTCTGATTTGTCCATT | |
| sdz-6-F | ACAATCGGGCGTTCAATTC | |
| sdz-6-R | TCTGATAGCTGGCTGAGTGG | |
| **Chemicals** | **Source** | **Catalog number** |
| Mitotracker Deep Red FM | Molecular Probes, Invitrogen | M22426 |
| Bortezomib | Selleck Chemicals | S1013 |
| Levamisole | Sigma | 196142 |
| Trizol | Invitrogen | 15596026 |
| Q5® Site-Directed Mutagenesis Kit | NEB | E0554 |
| NEBuilder HiFi DNA Assembly Cloning Kit | NEB | E5520 |
| Trimethylpsoralen | Thermo Fisher | 229881000 |
| iScript cDNA synthesis kits | Bio-Rad | 1708891 |
| PerfeCTa® SYBR® Green FastMix®, Low ROX™ | VWR | 733-1390 P |
| **Software** | **Source** | |
| GraphPad Prism 8 | https://www.graphpad.com/ | |
| Fiji | https://imagej.net/imagej-wiki-static/Fiji | |

## Methods and protocols

### C. elegans strains and maintenance

*C. elegans* were maintained on nematode growth medium (NGM) plates seeded with OP50 *Escherichia coli*. Animals were maintained at 20 °C unless otherwise noted. See the Reagents and tools table for a list of all *C. elegans* strains used in this study.

## Synchronization of *C. elegans*

To age-synchronize animals, gravid adult *C. elegans* were washed off plates with M9 buffer and transferred into a 5 mL conical tube. Clear pellets were resuspended in 2 mL of M9 and 1 mL of bleaching solution (600 uL of sodium hypochlorite solution, 160 uL of 5 M NaOH, and 240 uL of $H_2O$). Released embryos were washed twice with 5 mL of M9 and resuspended in a final volume of 3 mL of M9. Embryos were incubated at 20 °C with continual rotation for 20–24 h to hatch synchronized L1s.

## *C. elegans* RNAi

RNA interference was achieved by bacterial feeding using clones from the MRC RNAi library, after verifying sequence identity. All RNAi bacteria were not diluted, with one exception. *ubq-2(RNAi)* can cause developmental arrest; therefore, bacteria were diluted 10-fold with either empty vector RNAi or RNAi indicated within the text. Synchronized L1 were fed with an empty vector at 20 °C for 20 h, then transferred to diluted *ubq-2(RNAi)* to prevent developmental arrest.

## Orsay virus preparation

The *C. elegans rde-1* strain was infected by Orsay virus filtrate and maintained at 20 °C. The stably infected plates were grown to a mixed stage, washed off with S-medium, and then added to the S-medium with concentrated OP50 *E. coli*. The flask was then maintained at 160 rpm in a 20 °C shaking incubator for 7 days. Orsay virus was filtered through a 0.22-um filter, and stored at −80 °C. Similar levels of viral titer to stocks obtained from the Troemel laboratory were confirmed via RT-qPCR (*RNA1, pals-5*) and fluorescence (*pals-5p::GFP*).

## Orsay virus infection

For all strains, except those with *ulp-4(tm3688)*, synchronized L1 larvae were put onto 6 cm RNAi plates and incubated at 16 °C until the L4 stage, then a 200 uL of 1:1 dilution of Orsay Virus was top added onto the plates, which were dried at room temperature. Infected worms were incubated at 20 °C until collection. For AVS876-878 and AVS911-912, synchronized L1 larvae on 10 cm empty vector RNAi plates were incubated at 16 °C, then at L4, ~100 *ulp-4(tm3688)* homozygotic animals, with or without transgenes, were moved to 6 cm plates for infection. For the comparison of the induction of antiviral response between young and old animals, the old cohort was synchronized 3 days before the young cohort: ~300 animals were in each cohort. Animals were fed with empty vector RNAi and maintained at 20 °C prior to treatment with the virus. For viral infection, either L4 stage animals (day = 0) or animals at day 3 of adulthood were simultaneously treated with 200 ul of a 1:1 dilution of Orsay virus. Infected animals were incubated at 20 °C for 24 h.

## Bortezomib treatment

Synchronized L1 were plated onto 6 cm HT115 empty vector plates and incubated at 16 °C until L4. A 10 uM stock solution of bortezomib (Selleck Chemicals), resuspended in dimethyl sulfoxide (DMSO), was mixed with M9 and top-plated onto plates for a final concentration of 2.5 uM bortezomib per plate. Control plates were top-plated with an equal amount of DMSO in M9 buffer. Plates were then incubated at 20 °C for 24 h prior to collection in TRIZOL.

## Prolonged heat stress

Synchronized L1 were plated onto 6 cm HT115 empty vector plates and incubated at 16 °C until L4. Experimental plates were then incubated at 28 °C for 24 h, while control plates were incubated at 20 °C. Animals were collected in TRIZOL after 24 h. For the comparison of the induction of *pals-5* between young and old animal cohorts after heat stress, the cohort of old animals were age-synchronized 3 days before the young cohort; ~300 animals were in each cohort, and animals were maintained on empty vector RNAi at 20 °C prior to prolonged heat stress. Each cohort were simultaneously incubated at 28 °C for 24 h, while controls were maintained at 20 °C. Animals were collected in TRIZOL immediately after treatment.

## Molecular biology and transgenesis

To get a fusion construct containing the *ulp-4* coding sequence, the GFP fragment of the pPD95.75 vector was replaced by 4736 bp of the *ulp-4* genomic sequence fragment starting from the start codon of the *ulp-4* open reading frame. For intestinal expression, *ges-1p::ulp-4* was generated by PCR-cloning the 2073 bp *ges-1* promoter fragment upstream of *ulp-4* coding sequence. For epidermal expression, *ges-1p* was replaced by 803 bp of the *dpy-7* promoter. For expression in muscle, *ges-1p* was replaced by 2681 bp of the *myo-3* promoter. For neuronal expression, *ges-1p* was replaced by 1601 bp of the *rab-3* promoter. Tissue-specific ULP-4 expression plasmids were then injected into AVS876 at 25 ng/uL; *myo-2p::mCherry* was used as a co-injection marker.

The *rpl-28p::mScarlet::drh-1* plasmid was a gift from the laboratory of Gary Ruvkun (MGH/HMS). NEB Q5® Site-Directed Mutagenesis Kit (E0554) was used to generate the *rpl-28p::mScarlet::drh-1(K647A + K731A)* construct. *rpl-28p::mScarlet::drh-1* and *rpl-28p::mScarlet::drh-1(K647A + K731A)* were injected into AVS882 at 25 ng/uL; *myo-2p::GFP* was used as co-injection marker.

To construct *ges-1p::6xHIS::drh-1*, the *ulp-4* genomic sequence of *ges-1p::ulp-4* was replaced by the *drh-1* genomic sequence with 6XHIS tagged at the N terminus. The NBE Q5® Site-Directed Mutagenesis Kit (E0554) was used to generate *ges-1p::6xHIS::drh-1(K647A + K731A)*; this construct or *ges-1P::6xHIS::drh-1* were injected into AVS882 at 10 ng/uL; *myo-2p::GFP* was used as a co-injection marker.

Constructs were generated with the NEBuilder HiFi DNA Assembly Cloning Kit (NEB #E5520). All fragments obtained by PCR amplification were confirmed by sequencing.

## TMP/UV Integration

L2-L4 stage transgenic animals expressing an extrachromosomal array were collected in M9, washed, and resuspended in 200 uL of M9. TMP solution was added to obtain a final concentration of 100 ug/mL, and animals were incubated at RT for 30 min in the dark. Animals were then transferred to an unseeded 100 mm plate

and exposed to 350 uJ (x100) long-wave UV in a Stratalinker 1800. F2 animals with a 90–100% expression of a transgene in F3 progeny were putative homozygotic integrated transgenic lines, which was confirmed and then 6x backcrossed to N2.

## RNA isolation and RT-qPCR

Animals were isolated with M9, washed until clear, 200 ul TRIZOL was added to the pellet and frozen at −80 °C. RNA was extracted using TRIZOL reagent according to the manufacturer's instructions. cDNA was prepared from total RNA using iScript (Bio-Rad) cDNA synthesis kits. RT-qPCR was performed using PerfeCTa® SYBR® Green FastMix®, Low ROX™ (VWR) on a Quantum studio 5 system. Each replicate was measured in triplicate. All gene expression was normalized to *snb-1* and *cdc-42* expression. RT-qPCR primer sequences are listed in the Reagents and tools table.

## Fluorescence microscopy of *C. elegans*

Animals were added to a 10 mM levamisole droplet on 2% agarose pads. Whenever possible, samples were randomized, scoring was blinded, and image acquisition was representative of the experimental cohort. Images were captured with the Leica DMi8 confocal microscope or Zeiss Axio Imager.M2m and processed with Fiji.

## Immunoblotting

Animals raised under each described condition were isolated with M9 and washed until the supernatant was clear. 2X SDS Laemmli buffer (4% SDS, 20% glycerol, 10% 2-mercaptoethanol, 0.004% bromophenol blue, 0.125 M tris-HCl, pH 6.8) was used to resuspend the pellet. Samples were boiled at 97 °C for 10 min. Normalized protein levels were loaded onto 7% SDS-PAGE and transferred onto Nitrocellulose membranes (Bio-Rad). After blocking with 5% non-fat milk, the membrane was probed with the designated primary and secondary antibodies (rabbit polyclonal anti-RFP, Rockland # 600-401-379; rabbit polyclonal anti-actin, CST #4967; Goat anti-Rabbit IgG (H + L) Secondary Antibody, Thermo Fisher # 31460), and developed with SuperSignal™ West Atto Ultimate Sensitivity Substrate (Thermo Fisher # A38555), and visualized by Bio-Rad ChemiDoc Imaging System. Analysis was performed using Fiji.

## Quantification and statistical analysis

Relative *pals-5p::GFP* and *mScarlet::DRH-1* expression was analyzed by Fiji. Background mean fluorescence intensity was subtracted from each animal.

Immunoblot quantification was analyzed using Fiji.

All statistical analysis was performed with GraphPad Prism. A two-tailed *t*-test was used to calculate *P* values; $*P < 0.05$. $**P < 0.01$, $***P < 0.001$, and $****P < 0.0001$.

## Data availability

No datasets were generated in this study that would require deposition into a public database.

The source data of this paper are collected in the following database record: biostudies:S-SCDT-10_1038-S44319-025-00589-0.

## Peer review information

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

## Acknowledgements

We would like to thank members of the Samuelson laboratory for their insight and assistance related to this project. We would like to thank the laboratories of Drs. Emily Troemel (UCSD), David Wang (WUSTL), and Gary Ruvkun (MGH/HMS) for reagents. Some strains were provided by the CGC, which is funded by NIH Office of Research Infrastructure Programs (P40 OD010440), or by NBP-Japan. We thank Dr. Maureen Ferran (RIT) for discussions and advice. We thank Dr. Mary Wines-Samuelson for the critical reading of this manuscript. We thank members of the Department of Biomedical Genetics (URMC) and the Western New York Worm Group for discussions, particularly Drs. Doug Portman, Benoit Biteau, Andrew Wojtovich, and Keith Nehrke (URMC). AVS would like to thank the individuals who supported his independent research program at URMC, especially Drs. Hartmut Land and Dirk Bohmann. AVS would also like to thank the many members of the *C. elegans* community for their advice, encouragement, support, and friendship

over the past two decades. Research reported in this publication was supported by the National Institute on Aging and the National Institute of General Medical Sciences of the National Institutes of Health under Award Numbers R21AG064519, RF1AG062593, and R21GM148859, respectively. The content is solely the responsibility of the authors and does not necessarily represent the official views of the National Institutes of Health.

## Author contributions

**Yun Zhang**: Conceptualization; Data curation; Formal analysis; Validation; Investigation; Visualization; Methodology; Writing—original draft; Writing—review and editing. **Andrew V Samuelson**: Conceptualization; Supervision; Funding acquisition; Investigation; Writing—original draft; Project administration; Writing—review and editing.

Source data underlying figure panels in this paper may have individual authorship assigned. Where available, figure panel/source data authorship is listed in the following database record: biostudies:S-SCDT-10_1038-S44319-025-00589-0.

## Disclosure and competing interests statement

The authors declare no competing interests.

# Expanded View Figures

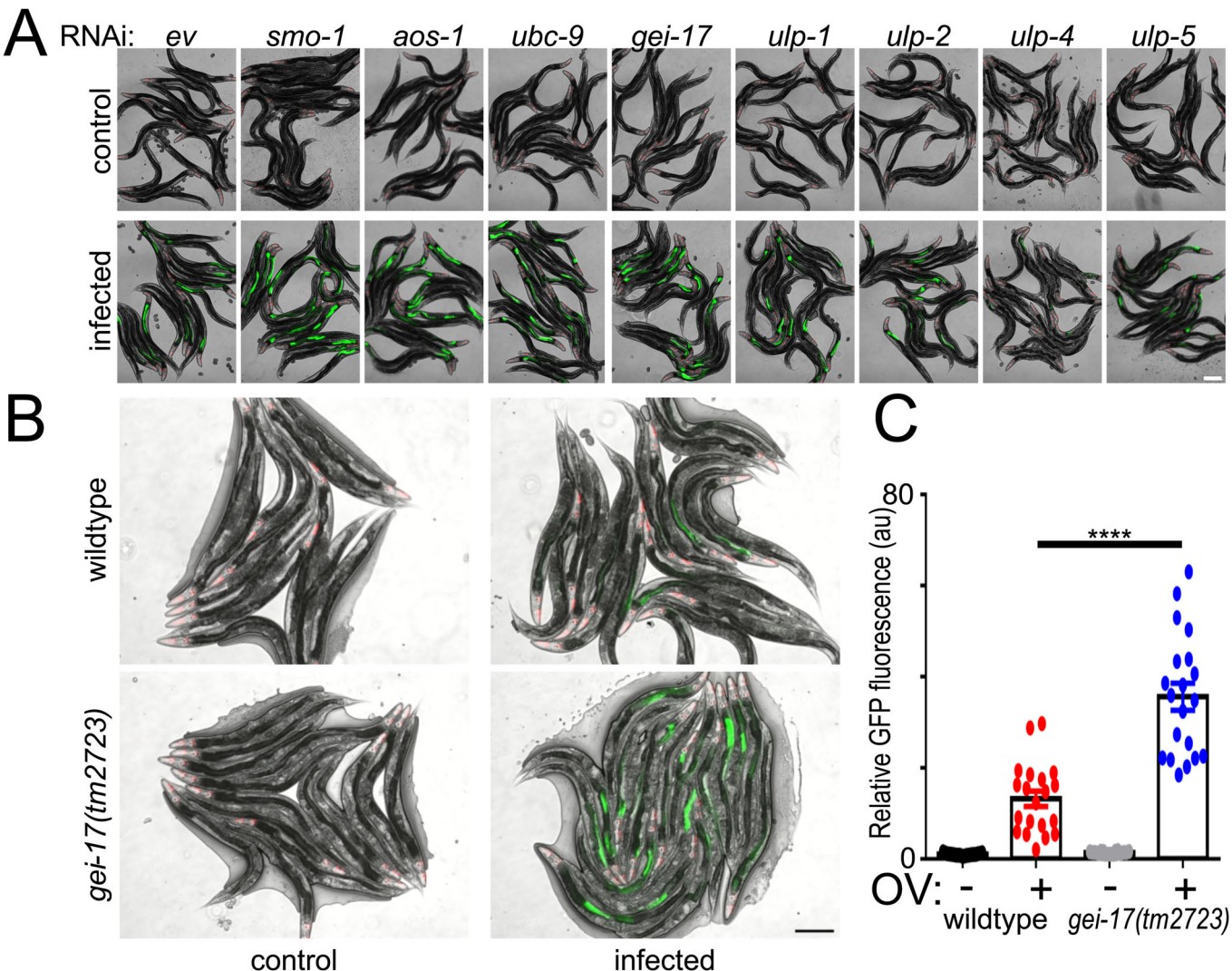

**Figure EV1. Inhibition of SUMOylation amplifies IPR induction after viral infection.**

(A) Representative images of *pals-5p::GFP* expression after RNAi of the indicated component of the SUMOylation machinery +/− viral infection. Quantification of GFP fluorescence is in Fig. 1B. Images of the empty vector (ev) and *ulp-4(RNAi)* are also shown in Fig. 1A. (B) Representative images of *pals-5p::GFP* expression in infected wildtype and *gei-17(tm2723)* null mutant animals. Scale bar = 200 μm. (C) Quantification of GFP fluorescence intensity of samples in (B). ****P < 0.0001. In all cases, values are the mean of 20 animals per trial, across three independent biological trials; error bars represent the SEM. A two-tailed *t*-test was used to calculate *P* value.

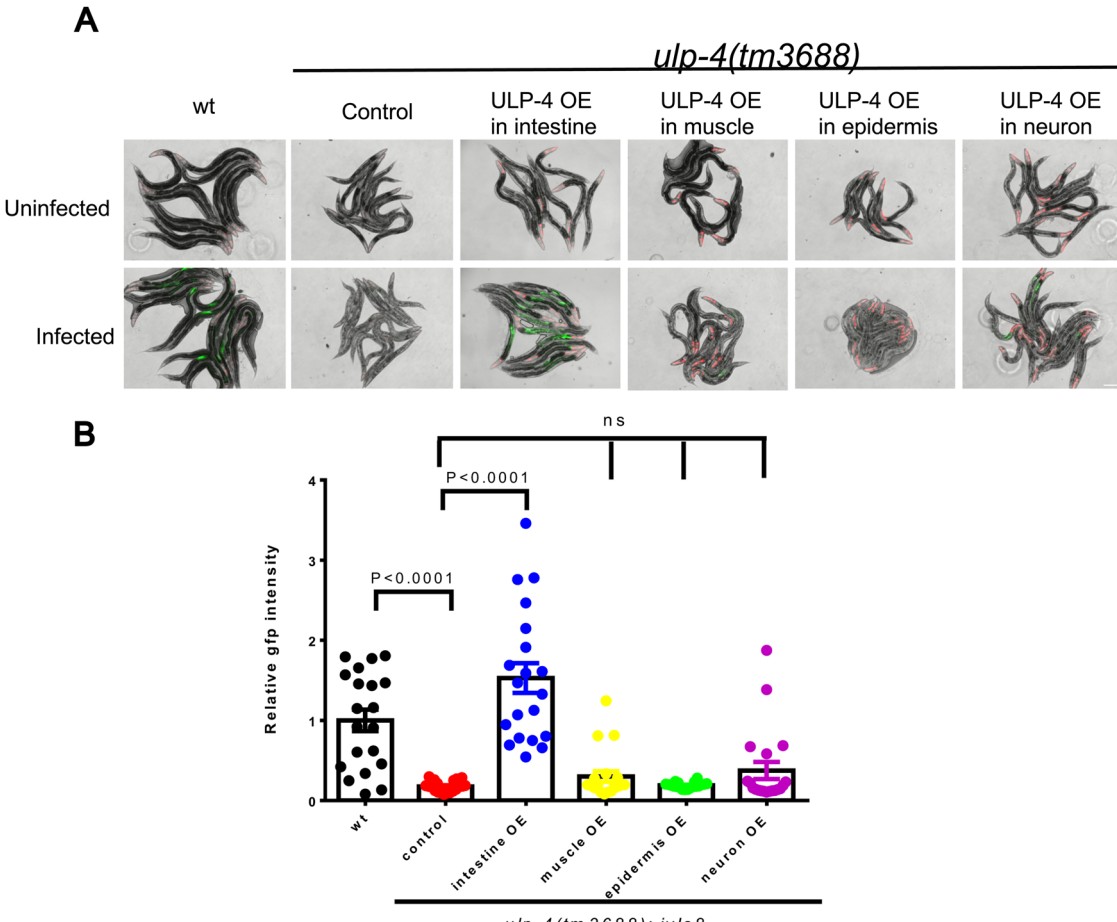

**Figure EV2. ULP-4 does not act outside of the intestine to regulate IPR following viral infection.**

(A) Representative images of *pals-5p::GFP* expression +/− viral infection of animals that were: wildtype, *ulp-4(tm3688)*, *ulp-4(tm3688);artEx95(ges-1p::ULP-4)*, *ulp-4(tm3688);artEx96(myo-3p::ULP-4)*, *ulp-4(tm3688);artEx104(dpy-7p::ULP-4)*, or *ulp-4(tm3688);artEx103(rab-3p::ULP-4)*, which restored ULP-4 expression within the intestine, body wall muscle, epidermis, or nervous system, respectively. Scale bar = 200 μm. (B) RT-qPCR analysis of endogenous *pals-5* mRNA of samples in (A). P values are provided within the panel. In all cases, values are the mean of 20 animals per trial, across three independent biological trials; error bars are SEM. A two-tailed *t*-test was used to calculate P values.

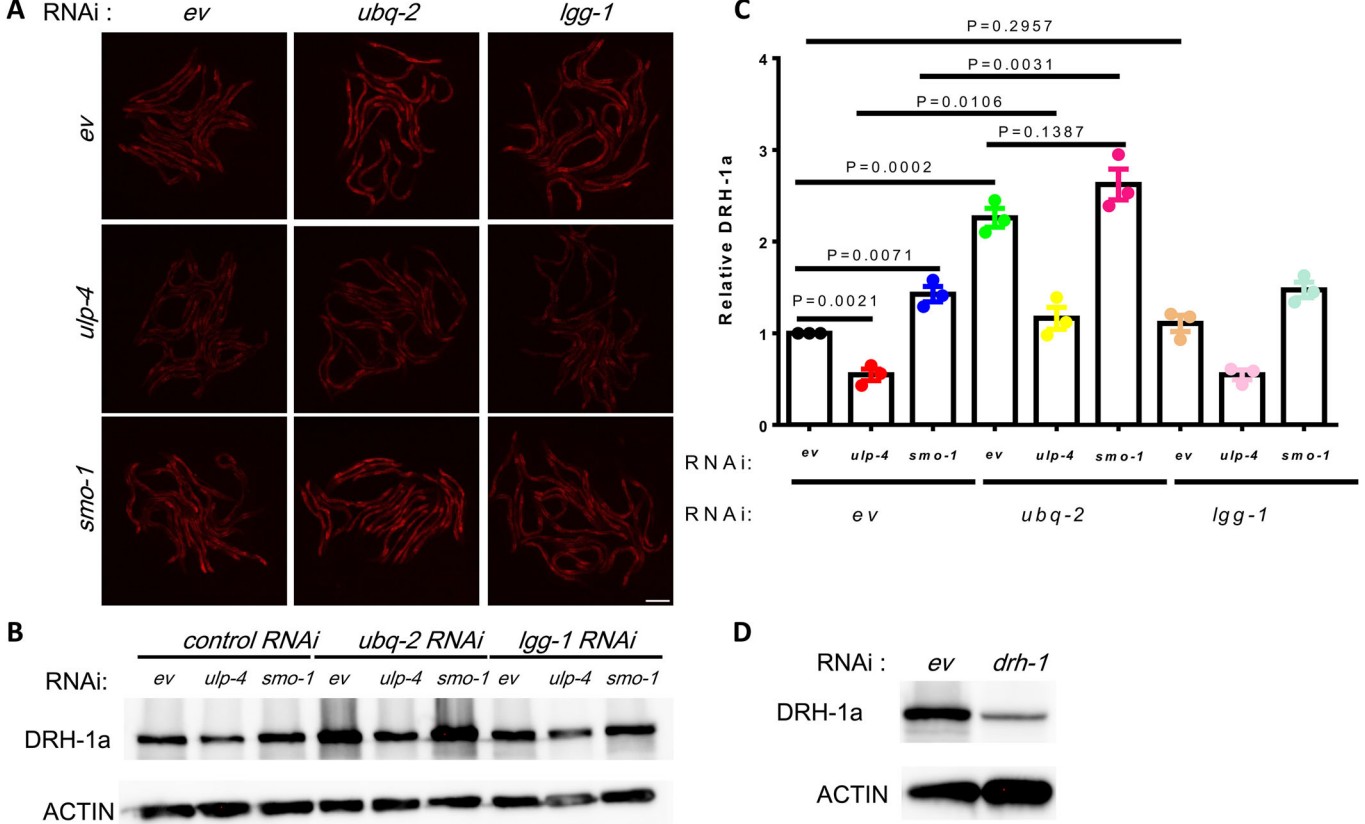

**Figure EV3.  DRH-1 is degraded through proteasome.**

(A) Representative images of *mScarlet::DRH-1* expression after RNAi treatment with: empty vector, *ubq-2, lgg-1, ulp-4, ulp-4+ubq-2, ulp-4+lgg-1, smo-1, smo-1+ubq-2,* or *smo-1+lgg-1,* respectively. Scale bar = 200 μm. (B) Representative immunoblot of DRH-1 levels. (C) Quantification of DRH-1 protein levels. *P* values are provided within the panel. (D) Representative immunoblot of DRH-1 levels after empty vector or *drh-1(RNAi)*. In all cases, values are the mean of 20 animals per trial, across three independent biological trials; error bars are SEM. A two-tailed *t*-test was used to calculate *P* values.

A

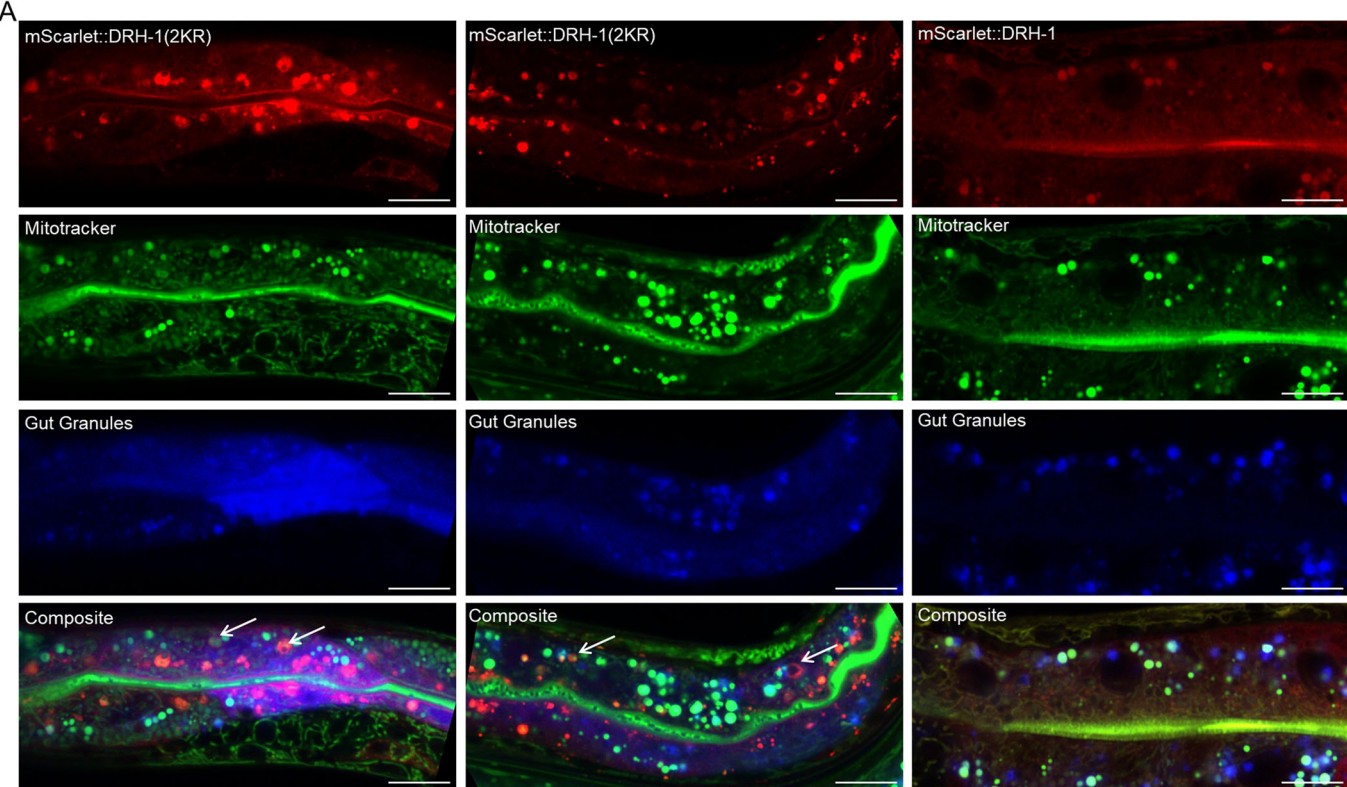

**Figure EV4.  NonSUMOylated DRH-1 translocates to the mitochondria.**

(**A**) Representative images of an intestinal cell of an animal expressing either *mScarlet::DRH-1 or mScarlet::DRH-1(2KR)*. Far Red Mitotracker was used to mark mitochondria, lysosomal-related organelles/gut granules are indicated (405-nm blue channel). Arrows highlight NonSUMOylated DRH-1 localization to the mitochondria periphery. Scale bar = 10 µm.

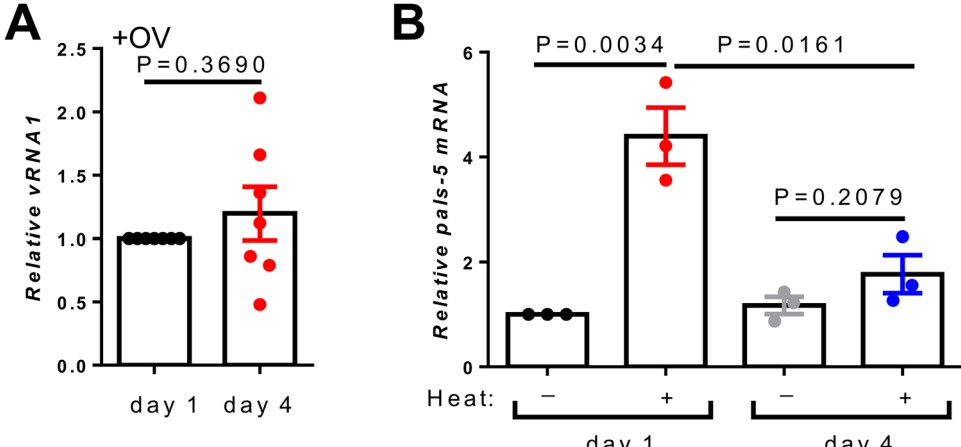

**Figure EV5. The IPR induced by a non-viral trigger declines during aging.**

(**A**) RT-qPCR analysis of viral *RNA1* levels in day 1 or day 4 animals. Viral infection was initiated at L4 or day 3 of adulthood, and animals were collected 24 h later. (**B**) RT-qPCR analysis of endogenous *pals-5* induction in day 1 or day 4 animals treated with prolonged heat stress. Heat stress was initiated at L4 or day 3 of adulthood at 28 °C for 24 h. Control plates were maintained at 20 °C. In all cases, values are the mean across three independent biological trials; error bars are SEM. A two-tailed *t*-test was used to calculate *P* values, which are provided within the panels.

