## [Peer Review File · EMBO Reports]

Aging impairs the antiviral defense in *Caenorhabditis elegans* due to loss of DRH-1/RIG-I deSUMOylation by ULP-4/SEN7

Yun Zhang and Andrew Samuelson

Corresponding author(s): Andrew Samuelson (andrew_samuelson@urmc.rochester.edu)

Review Timeline:

Submission Date:	3rd Dec 24
Editorial Decision:	23rd Dec 24
Revision Received:	6th Jun 25
Editorial Decision:	23rd Jul 25
Revision Received:	27th Aug 25
Accepted:	15th Sep 25

Transaction Report:

Dear Dr. Samuelson

Thank you for the submission of your research manuscript to our journal. Three referees agreed to review your manuscript. So far, we have received two referee reports that are copied below. Given that both referees are in fair agreement that you should be given a chance to revise the manuscript, I would like to ask you to begin revising your study along the lines suggested by the referees.

Please note that this is a preliminary decision made in the interest of time, and that it is subject to change should the third referee offer very strong and convincing reasons for this. As soon as we receive the final report on your manuscript, we will forward it to you as well.

Please address all referee concerns in a complete point-by-point response. Acceptance of the manuscript will depend on a positive outcome of a second round of review. It is EMBO Reports policy to allow a single round of revision only and acceptance or rejection of the manuscript will therefore depend on the completeness of your responses included in the next, final version of the manuscript.

We realize that it is difficult to revise to a specific deadline. In the interest of protecting the conceptual advance provided by the work, we recommend a revision within 3 months (March 23rd, 2025). Please discuss the revision progress ahead of this time with the editor if you require more time to complete the revisions.

I am also happy to discuss the revision further via e-mail or a video call, if you wish.

*******IMPORTANT NOTE:**

We perform an initial quality control of all revised manuscripts before re-review. Your manuscript will FAIL this control and the handling will be delayed IN CASE the following APPLIES:

- 1) A data availability section providing access to data deposited in public databases is missing. If you have not deposited any data, please add a sentence to the data availability section that explains that.
- 2) Your manuscript contains statistics and error bars based on $n=2$. Please use scatter blots in these cases. No statistics should be calculated if $n=2$.

When submitting your revised manuscript, please carefully review the instructions that follow below. Failure to include requested items will delay the evaluation of your revision. *****

- 1) a .docx formatted version of the manuscript text (including legends for main figures, EV figures and tables). Please make sure that the changes are highlighted to be clearly visible.
- 2) individual production quality figure files as .eps, .tif, .jpg (one file per figure). Please download our Figure Preparation Guidelines (figure preparation pdf) from our Author Guidelines pages <https://www.embopress.org/page/journal/14693178/authorguide> for more info on how to prepare your figures.
- 3) a .docx formatted letter INCLUDING the reviewers' reports and your detailed point-by-point responses to their comments. As part of the EMBO Press transparent editorial process, the point-by-point response is part of the Review Process File (RPF), which will be published alongside your paper.
- 4) a complete author checklist, which you can download from our author guidelines (<<https://www.embopress.org/page/journal/14693178/authorguide>>). Please insert information in the checklist that is also reflected in the manuscript. The completed author checklist will also be part of the RPF.
- 5) Please note that all corresponding authors are required to supply an ORCID ID for their name upon submission of a revised manuscript (<<https://orcid.org/>>). Please find instructions on how to link your ORCID ID to your account in our manuscript tracking system in our Author guidelines (<<https://www.embopress.org/page/journal/14693178/authorguide#authorshipguidelines>>)
- 6) We replaced Supplementary Information with Expanded View (EV) Figures and Tables that are collapsible/expandable online.

A maximum of 5 EV Figures can be typeset. EV Figures should be cited as 'Figure EV1, Figure EV2' etc... in the text and their respective legends should be included in the main text after the legends of regular figures.

7) Please note that a Data Availability section at the end of Materials and Methods is now mandatory. In case you have no data that requires deposition in a public database, please state so instead of refereeing to the database. See also < <https://www.embopress.org/page/journal/14693178/authorguide#dataavailability>>. Please note that the Data Availability Section is restricted to new primary data that are part of this study.

Additional information on source data and instruction on how to label the files are available <<https://www.embopress.org/page/journal/14693178/authorguide#sourcedata>>.

10) Figure legends and data quantification:

- the name of the statistical test used to generate error bars and P values,
 - the number (n) of independent experiments (please specify technical or biological replicates) underlying each data point,
 - the nature of the bars and error bars (s.d., s.e.m.)
- If the data are obtained from n {less than or equal to} 5, show the individual data points in addition to the SD or SEM.
- If the data are obtained from n {less than or equal to} 2, use scatter blots showing the individual data points.

11) Our journal encourages inclusion of *data citations in the reference list* to directly cite datasets that were re-used and obtained from public databases. Data citations in the article text are distinct from normal bibliographical citations and should directly link to the database records from which the data can be accessed. In the main text, data citations are formatted as follows: "Data ref: Smith et al, 2001" or "Data ref: NCBI Sequence Read Archive PRJNA342805, 2017". In the Reference list, data citations must be labeled with "[DATASET]". A data reference must provide the database name, accession number/identifiers and a resolvable link to the landing page from which the data can be accessed at the end of the reference. Further instructions are available at <<https://www.embopress.org/page/journal/14693178/authorguide#referencesformat>>.

12) All Materials and Methods need to be described in the main text using our 'Structured Methods' format. According to this format, the Methods section includes a Reagents and Tools Table (listing key reagents, experimental models, software and relevant equipment and including their sources and relevant identifiers) followed by a Methods and Protocols section describing the methods, ideally using a step-by-step protocol format. The aim is to facilitate adoption of the methodologies across labs. Please download and fill our Reagents and Tools Table template (.docx), which you can find in our author guidelines: <https://www.embopress.org/page/journal/14693178/authorguide#structuredmethods>.

13) As part of the EMBO publication's Transparent Editorial Process, EMBO Reports publishes online a Review Process File to accompany accepted manuscripts. This File will be published in conjunction with your paper and will include the referee reports, your point-by-point response and all pertinent correspondence relating to the manuscript.

Yours sincerely,

=====

Referee #1:

In this study, Zhang and Samuelson investigated the role of SUMOylation in regulating antiviral innate immunity in *C. elegans*. They observed that inhibition of the SUMOylation machinery led to an increase in the expression of *pals-5p::GFP*, a reporter for Orsay virus infection. In contrast, inhibition of the SUMO peptidase ULP-4 resulted in reduced expression of *pals-5p::GFP* under conditions of Orsay virus infection. The authors further examined the potential interaction between the SUMOylation pathway and the known antiviral response pathway in *C. elegans* involving DRH-1. They propose that ULP-4 regulates antiviral immunity by deSUMOylating DRH-1.

While this research explores an intriguing area, the conclusions presented in the manuscript rely heavily on indirect evidence. Furthermore, the manuscript lacks quantitative data in several critical sections, as outlined below. The authors should address the following points:

1. A major limitation of the study is the absence of direct evidence showing SUMOylation of DRH-1. The central conclusion that SUMOylation of DRH-1 suppresses antiviral immunity, while ULP-4-mediated deSUMOylation activates it, remains speculative without this evidence. The authors should conduct experiments to demonstrate whether DRH-1 is SUMOylated. If SUMOylation is observed, they should also provide evidence that ULP-4 reduces the levels of SUMOylated DRH-1.

2. The conclusion in the final paragraph of the results, titled "Antiviral response declines during aging due to dysregulation of DRH-1 SUMOylation," is not supported by the data. In the SI Appendix, Fig. 9B, although knockdown of *smo-1* increases *pals-5p::GFP* expression at both day 1 and day 4 compared to the empty vector (EV) control, a significant decline in *pals-5p::GFP* expression is observed between day 1 and day 4 in the *smo-1* knockdown condition. This decline is comparable to the relative decrease observed in the EV control. Similarly, Fig. 7D demonstrates substantially lower *pals-5* mRNA levels in DRH-1(2KR) worms at day 4 compared to day 1, indicating that the decline in antiviral immunity with age may be independent of DRH-1 SUMOylation.

Additionally, day 1 *smo-1* RNAi samples are missing in the SI Appendix, Fig. 9D. To support their conclusion, the authors should quantify DRH-1a protein levels and conduct a relative analysis (from day 1 to day 4) for both EV and *smo-1* RNAi.

3. To confirm that ULP-4 acts specifically through the DRH-1 pathway, the authors should include controls from other antiviral pathways, such as RNAi. Given the marginal increase in viral RNA upon *ulp-4* knockdown (Fig. 3B), additional controls are necessary to strengthen the claim of specificity.

4. The authors state that "Loss of *ubq-2* was sufficient to significantly increase fluorescence of the mScarlet::DRH-1 transgene and DRH-1 protein levels in animals lacking *ulp-4*, *smo-1*, or that were otherwise wildtype". It is not clear how the authors concluded that the fluorescence and protein levels were significantly different, as no quantification of the data was carried out.

5. Quantification and statistical analysis for mScarlet::DRH-1 levels should be included throughout the manuscript (e.g., Fig. 3, Fig. 7, SI Appendix Fig. 4, SI Appendix Fig. 9). Similarly, relative levels of DRH-1a should be quantified in Fig. 7, SI Appendix Fig. 4, SI Appendix Fig. 8, and SI Appendix Fig. 9, along with appropriate statistical analysis.

6. Fig. 4B shows a very high overlap between mScarlet::DRH-1 and GFP::MITO. However, in Fig. 4C, the overlap between mScarlet::DRH-1 and GFP::MITO is not high. Therefore, quantification of the overlap between mScarlet::DRH-1 and mitochondria markers is essential to clarify the relationship.

7. In Fig. 4C, the authors show that the GFP::SMO-1 does not overlap with mScarlet::DRH-1. However, it is unclear whether

GFP::SMO-1 accurately represents SUMOylation. The authors should include controls (e.g., non-infection conditions) to demonstrate GFP::SMO-1 overlap with mScarlet::DRH-1.

8. In Fig. 6C, the authors should provide data on virus load and clearance in *ulp-4* mutants to strengthen their conclusions.

9. The authors should include statistical analyses for Figs. 2C, 3E, and 3F to support their findings.

10. Several figures contain text with red underlines, likely due to assembly in PowerPoint. The authors should correct these formatting issues for clarity and presentation quality.

Referee #2:

Report:

Zhang et al. investigate the posttranslational regulation of DRH-1 and present convincing evidence that the sumoylation state of DRH-1 influences its ability to regulate a key innate immune program in *C. elegans*, the Intracellular Pathogen Response (IPR). Since the discovery that DRH-1 acts as a genuine pattern recognition receptor that is orthologous to mammalian RIG-I (Sowa et al. 2020, PMID: 31619561), there has been considerable interest in mechanistic insights into how DRH-1 functions to induce antiviral immunity. Subsequent work showed changes in the localization of DRH-1 under viral infection (Batachari et al. 2024, PMID: 38980902), but molecular-level mechanistic understanding of the regulation of DRH-1 remained unclear.

Here, the authors perform a comprehensive RNAi screen of known sumoylation machinery in *C. elegans* and identify the SUMO isopeptidase ULP-4 as critical for induction of the IPR under Orsay virus-infected conditions. The authors nicely demonstrate that loss of *ulp-4* results in protein-level degradation of DRH-1 in the proteasome, suppressing its ability to induce the IPR. Zhang et al. mutate two predicted sumoylation sites on DRH-1 and demonstrate that this mutant form has the opposite effect and promotes IPR induction. Of particular note and impact for this manuscript is the demonstration that activated DRH-1 co-localizes with intestinal mitochondria (although I have a few questions about this analysis that the authors should clarify; see below). This result is exciting and of value to virologists because it demonstrates a conserved role for mitochondria as a key location for antiviral innate immune signaling despite no clear MAVS ortholog in *C. elegans*. This insight opens exciting new opportunities to identify novel mitochondrially localized signaling mechanisms that act downstream of DRH-1 in the future for direct comparison to RIG-I signaling in other systems. This study is also impactful, as it adds to a growing understanding of connections between the IPR and its regulation via sumoylation mechanisms (Tecele et al. 2024 BioRxiv: <https://doi.org/10.1101/2024.10.03.616425>)

Finally, the authors provide a link between nematode aging, the stability of DRH-1, and the downregulation of *ulp-4* expression at the transcriptional level. This is a fascinating result, but I see some areas where the authors may want to consider expanding this analysis, as noted below. I view this manuscript as suitable for EMBO Reports, but I do have some recommendations for additional experiments that would strengthen this study and should be considered before its acceptance for publication.

Major Comments:

-The manuscript's title, introduction, and discussion center on the theme of antiviral defenses declining with age. Figure 7 and supplemental figures show an age-associated decline in DRH-1 and suppression of the IPR with age. However, this analysis would be strengthened by demonstrating age-associated susceptibility to OV by either RNA1 qPCR and/or OV FISH. Without some analysis of viral load, one interpretation of reduced IPR induction in D4 adults could be a myriad of physiological effects in aged adult animals (reduced feeding, changes in intestinal integrity, etc.), resulting in lower viral load compared to D1 adults and thus lower IPR induction. Given the biochemical analysis performed on DRH-1, it's unlikely this is the case, but the demonstration that OV load is at least similar, or perhaps higher if aligned with other results presented, in D4 adults is an important control. Finally, I would recommend that the analysis of *ulp-4* mRNA downregulation be moved into the main Figure 7 instead of the supplement.

-Related to the above, more methodological details are needed for viral infections - how many worms per plate, length of time exposed to the virus, if D1 vs. D4 infections were performed in parallel (ie same infection start time) or if the same population of worms was used to stagger the infection timing across different days, etc. Similarly, some description of how OV preps were made and how many distinct OV lysate preps were used in the study would be of value. OV preps are notoriously variable from prep to prep, so it's essential to know if this study was all conducted from the same preparation or across multiple independent preps.

-The observation that DRH-1 during OV infection, and DRH-1(2KR) are mitochondrially localized are fascinating results. A few questions/comments related to the provided confocal images: i) For Figure 4B, it would be useful to demonstrate that puncta formation and co-staining with mitochondria is specific to OV-infected cells. As the authors note, usually, only a few intestinal cells per animal are actively infected. Adding OV FISH to these experiments and imaging conditions combined with comparison to uninfected cells within the same animal would add value to the result that mitochondrial localization of DRH-1 is an OV infection-specific event (imaging akin to Batachari et al. 2024 Fig. 5 but with the addition of the mito labeling); ii) I note a difference in DRH-1 localization relative to the mitochondria for mScarlet::DRH-1 in 4B (seems more internal/overlapping to the mitochondrial signal) versus mScarlet::DRH-1(2KR) in 5F,G and S7 (more ring-like around the mitochondria, as if on the OMM). The authors note that the expression of the two constructs is relatively similar, so this is likely not an expression-specific effect. Some clarification here, or more representative images of each strain, would help distinguish if this difference in localization is

meaningful or specific to the images selected for the manuscript.

-Given that the two predicted sumoylation sites affect DRH-1 localization and ability to activate the IPR, it should be relatively straightforward to test each site independently to assess the contribution of K647 and K731 by modifying existing constructs to have one or the other lysines retained. Especially if the K731 in the helicase domain is the essential driver of this effect, that would be valuable mechanistic information about DRH-1 function. Ideally, the 2KR analysis could also be done on endogenous DRH-1, but this is less critical for publication. If there is a reason why independent lysine analysis isn't feasible or was tested and inconclusive, the authors should articulate that.

-I'm curious about the specificity of DRH-1 as a target for ULP-4 vs. 1, 2, or 5. Is anything known about tissue-specific expression of the SUMO isopeptidases? It would be interesting to perform smFISH on each ulp to determine if ulp-4 is intestinal-specific while others have roles in other tissues and thus are unlikely to act on DRH-1. Or, conversely, could intestinal expression of ULP-1, 2, or 5 in an ulp-4(0) mutant display the same activity on DRH-1 and IPR regulation?

Minor comments:

-Introduction: When discussing IPR genes related to abiotic stress responses, it might be helpful for readers to include references to Panek et al. 2020 (PMID: 32193347) and Bardan Sarmiento et al. (PMID: 38795346), as these provide the best described mechanistic understanding of IPR genes related to the proteotoxic stress response.

-Results "NonSUMOylated DRH-1 translocates to mitochondria upon viral infection." There is a genotype ~2/3 of the way into the paragraph listed as mScarlet::GFP, which I assume should be mScarlet::DRH-1.

-Suggested phrasing change - I don't believe sdz-6 is represented in the core ~80 IPR genes from Reddy et al. 2019 referenced in the introduction, but it is an Orsay infection-induced gene (Chang et al. 2017 PMID: 28415971); this distinction should be clarified.

-Related to Orsay-induced gene expression, the authors might consider commenting on ulp-4 expression from existing RNAseq datasets in the context of OV infection or heat shock-induced RNAi. If ulp-4 is not virus-induced, how might the balance between SUMOylated vs deSUMOylated DRH-1 be regulated? Are there other PTMs predicted for DRH-1?

-Fig 1A appears to be missing the scale bar described in the legend.

-Fig 4C and Fig 5G the legends could more directly describe arrows added to overlay images and what specific localization the authors are highlighting.

-Figure 5: Typos: i) capitalized jyls8, ii) there appears to be an issue with the nomenclature where E figure panel label is between mScarlet::DRH-1(WT); GFP::mito.

-The authors might consider briefly mentioning RNA+ viruses that infect other Caenorhabditis species (Santuil, Le Blanc, etc.) and cite associated works in this area.

-The authors might consider testing the heat shock inducible RNA1 strain developed by the Wang lab to determine if similar effects of ulp-1 on IPR suppression can be reproduced by conditional expression of RNA1.

Detailed Response to the critique**Referee #1:**

In this study, Zhang and Samuelson investigated the role of SUMOylation in regulating antiviral innate immunity in *C. elegans*. They observed that inhibition of the SUMOylation machinery led to an increase in the expression of *pals-5p::GFP*, a reporter for Orsay virus infection. In contrast, inhibition of the SUMO peptidase ULP-4 resulted in reduced expression of *pals-5p::GFP* under conditions of Orsay virus infection. The authors further examined the potential interaction between the SUMOylation pathway and the known antiviral response pathway in *C. elegans* involving DRH-1. They propose that ULP-4 regulates antiviral immunity by deSUMOylating DRH-1.

While this research explores an intriguing area, the conclusions presented in the manuscript rely heavily on indirect evidence. Furthermore, the manuscript lacks quantitative data in several critical sections, as outlined below. The authors should address the following points:

We thank the reviewer for their suggestions to improve the rigor of our manuscript and conclusions. In our responses, we refer to Figures according to how they are presented within the revised manuscript.

1. A major limitation of the study is the absence of direct evidence showing SUMOylation of DRH-1. The central conclusion that SUMOylation of DRH-1 suppresses antiviral immunity, while ULP-4-mediated deSUMOylation activates it, remains speculative without this evidence. The authors should conduct experiments to demonstrate whether DRH-1 is SUMOylated. If SUMOylation is observed, they should also provide evidence that ULP-4 reduces the levels of SUMOylated DRH-1.

We thank the reviewer for their suggestion. We agree that directly demonstrating SUMOylation of DRH-1 would support our model. Despite repeated efforts we have been unable to directly observe DRH-1 SUMOylation. We believe this is a technical limitation; SUMOylation is a transient modification, only a few infected intestinal cells within the animal would have altered SUMOylation (out of the 959 somatic cells), and only a fraction of a target protein is SUMOylated at steady state. Based on the data we provide, SUMOylation of DRH-1 also leads to degradation through the proteasome. Because of these technical limitations, we do not believe it will be feasible to work out the experimental conditions that would be necessary to directly observe DRH-1. This is why we focused on alternative genetic approaches, such as mutating the predicted DRH-1 SUMO sites, and showed that lysines 647 and 731 are the main SUMO sites as their mutation bypassed the need for ULP-4 in regulating DRH-1. We respectfully disagree with the reviewer's assertion that our conclusion is speculative. While indirect, we believe this finding, in conjunction with the additional genetic, molecular, and cellular analysis, are collectively sufficient to support our model. Nevertheless, we have softened our conclusion and added to the discussion

that we were unable to directly observe DRH-1 SUMOylation.

2. The conclusion in the final paragraph of the results, titled "Antiviral response declines during aging due to dysregulation of DRH-1 SUMOylation," is not supported by the data.

*We thank the reviewer for their analysis. We never intended to imply that increased SUMOylation of DRH-1 was the only age-associated change that diminishes the inducibility of the IPR. The age-associated loss in inducibility of a myriad of stress response genes is well known to occur via epigenetic changes in chromatin over the first few days of adulthood (e.g., PMID: 26212459), which we further highlight in the revised manuscript. Indeed, in the revised manuscript we show that the inducibility of *pals-5* by *drh-1*-independent, non-viral triggers also declines during aging (Fig. EV5B). Please see response to points 3&4 below, which further address the larger concern. We have also changed the header to soften our conclusion.*

3. In the SI Appendix, Fig. 9B, although knockdown of *smo-1* increases *pals-5p::GFP* expression at both day 1 and day 4 compared to the empty vector (EV) control, a significant decline in *pals-5p::GFP* expression is observed between day 1 and day 4 in the *smo-1* knockdown condition. This decline is comparable to the relative decrease observed in the EV control.

*We thank the reviewer for their analysis. These results were included in the original manuscript as an approach complementary to DRH-1(2KR). We agree that by itself the data presented in the mentioned figure was rather weak, which is why we had relegated it to the supplemental results. There are technical and biological limitations: *smo-1(RNAi)* is only a partial knockdown and it is difficult to assess both how RNAi affects pools of free and conjugated SMO-1 protein, and the overall impact on organismal physiology as a whole. Because of these limitations we have removed this analysis from the revised manuscript. We thank the reviewer for highlighting extraneous data that is not necessary to support our overall conclusions and only serve to diminish the overall quality of our manuscript.*

4. Similarly, Fig. 7D demonstrates substantially lower *pals-5* mRNA levels in DRH-1(2KR) worms at day 4 compared to day 1, indicating that the decline in antiviral immunity with age may be independent of DRH-1 SUMOylation.

*We respectfully disagree with the reviewer's conclusion. We do agree that the maximal induction of *pals-5* is lower in day 4 animals compared to day 1. However, the relative induction supports the conclusion that intestinal expression of *drh-1(2KR)* at least partially rescues the age-associated loss of the IPR. The relative induction of *pals-5* between either day 1 wild-type animals (47.1x, Fig 7E column 2) or intestinal rescue of full-length *drh-1* (56.6x, Fig 7E column 4) compared to intestinal rescue with the *drh-1(2KR)* mutant isoform (137.4x, Fig 7E column 6), reveals that *drh-1(2KR)**

hyper-induces pals-5 by 2.4-fold (i.e., columns 6 versus 4). In contrast to day 1, induction at day 4 in wild-type animals or drh-1(0) rescued with full length drh-1 is only 6.4x and 8.3x (respectively; Fig 7E, columns 8&10), which is ~10-fold lower than at day 1. In contrast, animals expressing the drh-1(2KR) isoform induce pals-5 49.5-fold (column 12), which is a 6-fold comparative increase (i.e, 12 versus 10). This 6-fold increase in the older animals is significantly greater than the 2.4-fold observed in the younger animals (p=0.02) and consistent with age-associated rescue. As an aside, hyper-induction of pals-5 after mutating a repressive post-translational modification in DRH-1 is not unexpected. In these experiments intestinally expressed (ges-1p) drh-1 or drh-1(2KR) is not sufficient to induce the IPR in the absence of virus (Fig 7E, columns 3&5, respectively). When we use a pan-somatic promoter (rpl-28p), which produces even higher levels of expression, we observe strong induction of pals-5 by DRH-1(2KR) in the absence of virus (Fig 7G/H). GFP fluorescence actually increases in older animals expressing drh-1(2KR) to a greater extent than in younger animals. We have quantified these levels of GFP fluorescence in the revised manuscript (Fig 7I), and while levels of the IPR increase in the absence of external virus, expression of the drh-1(2KR) transgene is unaltered (Fig 7J). Collectively, these results not only support our conclusion that dysregulation of DRH-1 SUMOylation during aging limits the induction of antiviral defense, but also suggests that aging itself is sufficient to trigger IPR activation. We have added this to the discussion.

There are several possible interpretations for why intestinal expression of drh-1(2KR) only partially rescues the induction of the IPR. We believe our response to #2 (epigenetic changes) is a reasonable explanation. Indeed, pals-5 inducibility by heat shock, a non-viral trigger, also declines with age, which we have included in the revised results and discussion (Fig EV5B). Additionally, there is evidence that levels of IPR induction scales with viral loads; therefore, we tested whether overall lower levels of pals-5 induction could simply be the result of lower overall viral loads in the older cohort. This was not the case: we observed no significant difference in viral loads between cohorts infected at different ages (Fig EV5A).

5. Additionally, day 1 *smo-1* RNAi samples are missing in the SI Appendix, Fig. 9D. To support their conclusion, the authors should quantify DRH-1a protein levels and conduct a relative analysis (from day 1 to day 4) for both EV and *smo-1* RNAi.

We thank the reviewer again. We removed this supplemental figure in the revised manuscript (please see #3 above).

6. To confirm that ULP-4 acts specifically through the DRH-1 pathway, the authors should include controls from other antiviral pathways, such as RNAi. Given the marginal increase in viral RNA upon *ulp-4* knockdown (Fig. 3B), additional controls are necessary to strengthen the claim of specificity.

We thank the reviewer for the suggestion. We are not claiming that the antiviral

activity of ULP-4 only occurs through the regulation of DRH-1. Rather, our genetic data provided in Fig 3B supports the notion that ULP-4 and DRH-1 have overlapping antiviral functions, which can be observed through regulation of the IPR. This regulation is specific to antiviral defense, as ulp-4 is not required for pals-5 induction by non-viral triggers (Appendix FigS1). It is possible that ULP-4 removes SUMOylation from other targets in response to viral infection, which could also have a role in antiviral defense. We think this is an excellent area to follow-up in future studies, but is beyond the scope of the current study.

7. The authors state that "Loss of *ubq-2* was sufficient to significantly increase fluorescence of the mScarlet::DRH-1 transgene and DRH-1 protein levels in animals lacking *ulp-4*, *smo-1*, or that were otherwise wildtype". It is not clear how the authors concluded that the fluorescence and protein levels were significantly different, as no quantification of the data was carried out.

We thank the reviewer for catching this oversight. We have included quantification in the revised manuscript (Fig EV3C); the results of the quantification support our prior conclusion.

8. Quantification and statistical analysis for mScarlet::DRH-1 levels should be included throughout the manuscript (e.g., Fig. 3, Fig. 7, SI Appendix Fig. 4, SI Appendix Fig. 9). Similarly, relative levels of DRH-1a should be quantified in Fig. 7, SI Appendix Fig. 4, SI Appendix Fig. 8, and SI Appendix Fig. 9, along with appropriate statistical analysis.

We again thank the reviewer for catching our oversight. In the revised manuscript we have either included quantification (revised Figs 3, 7, EV3) or removed the aforementioned figures (original SI Appendix 8&9). It should be noted that for the mScarlet::DRH-1 immunoblots, we used an antibody that is specific to the mScarlet epitope. Thus, quantifying levels for both the immunoblot and fluorescence is largely redundant. We clarify this in the revised manuscript. The results of the quantification support our prior conclusions.

9. Fig. 4B shows a very high overlap between mScarlet::DRH-1 and GFP::MITO. However, in Fig. 4C, the overlap between mScarlet::DRH-1 and GFP::MITO is not high. Therefore, quantification of the overlap between mScarlet::DRH-1 and mitochondria markers is essential to clarify the relationship.

*We thank the reviewer for the suggestion, but it is unclear to us what additional biological insight would be gained by comparing the relative co-localization of DRH-1 at mitochondria when using a GFP::mito reporter or Mitotracker. Both approaches show DRH-1 colocalization at the mitochondria, which has not previously been demonstrated in *C. elegans*. While the DRH-1 ortholog, RIG-I, is well known to translocate to the mitochondria upon viral infection, whether or not a similar*

mechanism occurs in C. elegans was unclear. In mammals, RIG-I localization at mitochondria occurs through interaction with MAVs, but C. elegans lack a clear MAVs ortholog. Our results suggest that mitochondria may play a conserved role in antiviral defense and open the possibility that RIG-I or other mammalian RIG-I paralogs may localize to mitochondria independent of MAVs interactions. We have added this to the discussion.

10. In Fig. 4C, the authors show that the GFP::SMO-1 does not overlap with mScarlet::DRH-1. However, it is unclear whether GFP::SMO-1 accurately represents SUMOylation. The authors should include controls (e.g., non-infection conditions) to demonstrate GFP::SMO-1 overlap with mScarlet::DRH-1.

We thank the reviewer for this suggestion. In the revised manuscript we provide these images in (Appendix Fig S3). In the absence of viral infection, GFP::SMO-1 and mScarlet::DRH-1 are diffusely distributed in the cytosol, with good overlap at the periphery of the cell.

11. In Fig. 6C, the authors should provide data on virus load and clearance in *ulp-4* mutants to strengthen their conclusions.

*We thank the reviewer for this suggestion. In the absence of *ulp-4*, animals have higher viral loads and fail to clear the virus. In the revised manuscript we provide this data in (Fig 6C).*

12. The authors should include statistical analyses for Figs. 2C, 3E, and 3F to support their findings.

We thank the reviewer for catching this oversight. In the revised manuscript we provide a statistical analysis for the data within these figures.

13. Several figures contain text with red underlines, likely due to assembly in PowerPoint. The authors should correct these formatting issues for clarity and presentation quality.

We thank the reviewer for pointing this out. We have corrected this in the revised manuscript.

Referee #2:

Report:

Zhang et al. investigate the posttranslational regulation of DRH-1 and present convincing evidence that the sumoylation state of DRH-1 influences its ability to regulate a key innate immune program in *C. elegans*, the Intracellular Pathogen Response (IPR). Since the discovery that DRH-1 acts as a genuine pattern

recognition receptor that is orthologous to mammalian RIG-I (Sowa et al. 2020, PMID: 31619561), there has been considerable interest in mechanistic insights into how DRH-1 functions to induce antiviral immunity. Subsequent work showed changes in the localization of DRH-1 under viral infection (Batachari et al. 2024, PMID: 38980902), but molecular-level mechanistic understanding of the regulation of DRH-1 remained unclear.

Here, the authors perform a comprehensive RNAi screen of known sumoylation machinery in *C. elegans* and identify the SUMO isopeptidase ULP-4 as critical for induction of the IPR under Orsay virus-infected conditions. The authors nicely demonstrate that loss of *ulp-4* results in protein-level degradation of DRH-1 in the proteasome, suppressing its ability to induce the IPR. Zhang et al. mutate two predicted sumoylation sites on DRH-1 and demonstrate that this mutant form has the opposite effect and promotes IPR induction. Of particular note and impact for this manuscript is the demonstration that activated DRH-1 co-localizes with intestinal mitochondria (although I have a few questions about this analysis that the authors should clarify; see below). This result is exciting and of value to virologists because it demonstrates a conserved role for mitochondria as a key location for antiviral innate immune signaling despite no clear MAVS ortholog in *C. elegans*. This insight opens exciting new opportunities to identify novel mitochondrially localized signaling mechanisms that act downstream of DRH-1 in the future for direct comparison to RIG-I signaling in other systems. This study is also impactful, as it adds to a growing understanding of connections between the IPR and its regulation via sumoylation mechanisms (Tecele et al. 2024 BioRxiv: <https://doi.org/10.1101/2024.10.03.616425>)

Finally, the authors provide a link between nematode aging, the stability of DRH-1, and the downregulation of *ulp-4* expression at the transcriptional level. This is a fascinating result, but I see some areas where the authors may want to consider expanding this analysis, as noted below. I view this manuscript as suitable for EMBO Reports, but I do have some recommendations for additional experiments that would strengthen this study and should be considered before its acceptance for publication.

We thank the reviewer for their kind appreciation of our work and recognizing the significance.

Major Comments:

14. -The manuscript's title, introduction, and discussion center on the theme of antiviral defenses declining with age. Figure 7 and supplemental figures show an age-associated decline in DRH-1 and suppression of the IPR with age. However, this analysis would be strengthened by demonstrating age-associated susceptibility to OV by either RNA1 qPCR and/or OV FISH.

We thank the reviewer for this excellent suggestion. In the revised manuscript we

show that viral loads in young and older animals 24 hours after infection are unchanged (Fig EV5A).

15. Without some analysis of viral load, one interpretation of reduced IPR induction in D4 adults could be a myriad of physiological effects in aged adult animals (reduced feeding, changes in intestinal integrity, etc.), resulting in lower viral load compared to D1 adults and thus lower IPR induction. Given the biochemical analysis performed on DRH-1, it's unlikely this is the case, but the demonstration that OV load is at least similar, or perhaps higher if aligned with other results presented, in D4 adults is an important control. Finally, I would recommend that the analysis of *ulp-4* mRNA downregulation be moved into the main Figure 7 instead of the supplement.

We agree with the reviewer that quantifying viral loads in young and old animals is an important control. We find that viral levels are similar between the two cohorts (Fig EV5A). We thank the reviewer for the additional suggestion and have moved what was SI Fig 9e into the main text (Fig 7F in the revised manuscript).

16. -Related to the above, more methodological details are needed for viral infections - how many worms per plate, length of time exposed to the virus, if D1 vs. D4 infections were performed in parallel (i.e., same infection start time) or if the same population of worms was used to stagger the infection timing across different days, etc. Similarly, some description of how OV preps were made and how many distinct OV lysate preps were used in the study would be of value. OV preps are notoriously variable from prep to prep, so it's essential to know if this study was all conducted from the same preparation or across multiple independent preps.

We thank the reviewer and updated our methods section as requested in the revised manuscript.

17. -The observation that DRH-1 during OV infection, and DRH-1(2KR) are mitochondrially localized are fascinating results. A few questions/comments related to the provided confocal images: i) For Figure 4B, it would be useful to demonstrate that puncta formation and co-staining with mitochondria is specific to OV-infected cells. As the authors note, usually, only a few intestinal cells per animal are actively infected. Adding OV FISH to these experiments and imaging conditions combined with comparison to uninfected cells within the same animal would add value to the result that mitochondrial localization of DRH-1 is an OV infection-specific event (imaging akin to Batachari et al. 2024 Fig. 5 but with the addition of the mito labeling);

We observe a similar pattern of DRH-1 puncta formation in only a few cells following viral infection, which is consistent with the findings from the Troemel laboratory. We never see this pattern in uninfected animals. As the reviewer notes, the Troemel laboratory has nicely shown via FISH that the intestinal cells with DRH-1 puncta also have viral RNA. We agree that adding FISH to our analysis would be a nice addition,

but unfortunately, we lacked the resources to conduct the requested experiment.

18. ii) I note a difference in DRH-1 localization relative to the mitochondria for mScarlet::DRH-1 in 4B (seems more internal/overlapping to the mitochondrial signal) versus mScarlet::DRH-1(2KR) in 5F,G and S7 (more ring-like around the mitochondria, as if on the OMM). The authors note that the expression of the two constructs is relatively similar, so this is likely not an expression-specific effect. Some clarification here, or more representative images of each strain, would help distinguish if this difference in localization is meaningful or specific to the images selected for the manuscript.

We thank the reviewer for noting these differences; we have gone back and reanalyzed our images in greater detail. In Fig 4B we observe an accumulation of DRH-1 at the mitochondria following infection; it is worth noting colocalization is observed using mScarlet::DRH-1 and GFP::mito, the latter labels the mitochondrial matrix effectively directing fluorescence to the inner compartment of the mitochondria. The mScarlet::DRH-1 puncta are larger than the GFP::mito, which suggests DRH-1 may actually be surrounding the mitochondria at these locations. In Fig 4C we are observing co-localization after viral infection using mScarlet::DRH-1 with MitoTracker Far Red (false colored to white); the latter typically labels throughout mitochondria and there we observe peripheral mitochondrial localization more clearly (white arrows) and less total overlap. Mitotracker is also specific to active mitochondria. Collectively, it is tempting to speculate that mitochondrial regions of total overlap with DRH-1 are inactive, perhaps due to viral accumulation at that location and are in the process of being removed (e.g., via mitophagy or fission). It is an intriguing possibility worth following up in subsequent studies!

In the current study, we limit ourselves to a more conservative conclusion that nonSUMOylated DRH-1 accumulates at the OMM after viral infection. In Fig 5F,G & Appendix FigS5, images are taken in the absence of infection and we clearly see the accumulation of mScarlet::DRH-1(2KR) at the mitochondrial periphery, which supports the role of SUMOylation in preventing mitochondrial localization of DRH-1. In our revised manuscript we have included additional images with mScarlet::DRH-1 with MitoTracker Far Red (false colored to the green channel) that support our conclusion (Fig EV4 in the revised manuscript); peripheral localization of DRH-1(2KR) occurs ~20% of the time. Lastly, it is worth noting that MitoTracker is a lipophilic dye, which also accumulates within lipid-rich gut granules. We find that DRH-1, but not DRH-1(2KR), also localized to gut granules, which produced an overlay consistent with internalization (as well as diffuse cytosolic staining).

19. -Given that the two predicted sumoylation sites affect DRH-1 localization and ability to activate the IPR, it should be relatively straightforward to test each site independently to assess the contribution of K647 and K731 by modifying existing constructs to have one or the other lysines retained. Especially if the K731 in the

helicase domain is the essential driver of this effect, that would be valuable mechanistic information about DRH-1 function. Ideally, the 2KR analysis could also be done on endogenous DRH-1, but this is less critical for publication. If there is a reason why independent lysine analysis isn't feasible or was tested and inconclusive, the authors should articulate that.

We thank the reviewer for the excellent suggestion. We attempted to do as requested to create transgenic lines overexpressing drh-1(K731R) throughout the soma. We found strong selective pressure against propagation of this transgene to subsequent generations: approximately half of the F1 animals underwent L1-L2 larval arrest, and all but one of ~20 F1 lines we created failed to transmit the extrachromosomal array. The one mScarlet::DRH-1(K731R) line we were able to generate had substantially lower levels of mScarlet expression than either the DRH-1(wt) or DRH-1(2KR) lines. Furthermore, expression of DRH-1(K731R) was still reduced with loss of ulp-4 in this one line, analogous to what we observed with wild-type DRH-1. Because of the preliminary and non-straightforward nature of these results, we decided to omit them from the revised manuscript but agree that a refined mutational analysis in a follow-up study could yield greater mechanistic insight.

20. -I'm curious about the specificity of DRH-1 as a target for ULP-4 vs. 1, 2, or 5. Is anything known about tissue-specific expression of the SUMO isopeptidases? It would be interesting to perform smFISH on each *ulp* to determine if *ulp-4* is intestinal-specific while others have roles in other tissues and thus are unlikely to act on DRH-1. Or, conversely, could intestinal expression of ULP-1, 2, or 5 in an *ulp-4(0)* mutant display the same activity on DRH-1 and IPR regulation?

*This is a great question! How specificity is obtained in deconjugating the various SUMO moieties in mammals or SMO-1 in *C. elegans* is a major unresolved question (outside of what we include in our discussion). Whether or when SMO-1 forms polymerize chains with itself or ubiquitin, like mammalian SUMO2/3, is unknown. It is also unknown why SMO-1 conjugation leads to distinct outcomes (degradation, translocation, altered protein-protein interactions, alteration in activity). This is an exciting area to follow-up!*

*While ULP-family members have distinct patterns of expression, it is unlikely that there is cross-reactivity between them; the deconjugating activity of mammalian SENP orthologs is specific (but general rules for how specificity is obtained have not yet emerged). ULP-4 is widely expressed and localizes to both the cytosol and mitochondria (PMID: 25187565). ULP-1 is also widely expressed throughout the soma and primarily localized within the nucleus (PMID: 31243051), which we have also observed using the syb366 fluorescently tagged allele of endogenous *ulp-1* (unpublished). In the case of ULP-1, we have unpublished findings that it is critical for other forms of stress response. ULP-2 has two splice isoforms that are expressed either in both the nucleus and cytosol or solely within the cytosol; and is primarily*

expressed within the hypodermis (tissues) (PMID: 26412237). ULP-5 is reported to be nuclear (wormbase.org), but the tissues of expression is unclear.

Minor comments:

-Introduction: When discussing IPR genes related to abiotic stress responses, it might be helpful for readers to include references to Panek et al. 2020 (PMID: 32193347) and Bardan Sarmiento et al. (PMID: 38795346), as these provide the best described mechanistic understanding of IPR genes related to the proteotoxic stress response.

We have added the suggested references to the introduction.

-Results "NonSUMOylated DRH-1 translocates to mitochondria upon viral infection." There is a genotype ~2/3 of the way into the paragraph listed as mScarlet::GFP, which I assume should be mScarlet::DRH-1.

Thank you for catching our mistake.

-Suggested phrasing change - I don't believe sdz-6 is represented in the core ~80 IPR genes from Reddy et al. 2019 referenced in the introduction, but it is an Orsay infection-induced gene (Chang et al. 2017 PMID: 28415971); this distinction should be clarified.

Fixed!

-Related to Orsay-induced gene expression, the authors might consider commenting on ulp-4 expression from existing RNAseq datasets in the context of OV infection or heat shock-induced RNAi. If ulp-4 is not virus-induced, how might the balance between SUMOylated vs deSUMOylated DRH-1 be regulated? Are there other PTMs predicted for DRH-1?

These are excellent suggestions and important gaps in knowledge worthy of follow-up! We touch upon this in our revised discussion.

-Fig 1A appears to be missing the scale bar described in the legend.

Fixed!

-Fig 4C and Fig 5G the legends could more directly describe arrows added to overlay images and what specific localization the authors are highlighting.

Added!

-Figure 5: Typos: i) capitalized jyls8, ii) there appears to be an issue with the

nomenclature where E figure panel label is between mScarlet::DRH-1(WT); GFP::mito.

Fixed!

-The authors might consider briefly mentioning RNA+ viruses that infect other Caenorhabditis species (Santuil, Le Blanc, etc.) and cite associated works in this area.

We thank the reviewer for making this suggestion. In the revised manuscript we have included this in the introduction. Furthermore, we have expanded upon the implications and the significance of using nematodes as a host-viral model.

-The authors might consider testing the heat shock inducible RNA1 strain developed by the Wang lab to determine if similar effects of ulp-1 on IPR suppression can be reproduced by conditional expression of RNA1.

We considered this, but decided against using an inducible system that uses heat shock, as heat shock also induces the IPR and we show that this induction is drh-1 independent.

Dear Dr. Samuelson

Thank you for the submission of your revised manuscript to EMBO reports. We have now received the full set of referee reports that is copied below.

As you will see, Referee #1 recommends publication without further revision. Referee #2 also supports publication after minor changes to text and figures.

Browsing through the manuscript myself, I noticed a few editorial things that we need before we can proceed with the official acceptance of your study.

1) Please provide up to 5 keywords on the title page.

2) Please update the 'Conflict of interest' paragraph to our new 'Disclosure and competing interests statement'. For more information see

<https://www.embopress.org/page/journal/14693178/authorguide#conflictsofinterest>

3) Please update the references to the alphabetical Harvard style. The abbreviation 'et al' should be used if there are more than 10 authors. You can download the respective EndNote file from our Guide to Authors

https://endnote.com/style_download/embo-reports/

4) The manuscript sections should be in the following order: Title page - Abstract & Keywords - Introduction - Results - Discussion - Methods - Data Availability - Acknowledgments - Disclosure Statement & Competing Interests - References - Figure Legends - (Main Tables with legends if applicable) - Expanded View Figure Legends.
- Materials and Methods should be just Methods

5) The Significance statement is not part of EMBO Reports articles and needs to be removed.

6) Please provide a legend for Dataset EV1. The legend needs to be in a separate tab of the .xls file.

7) The individual Appendix Figure files are not required, the merged Appendix PDF is sufficient. But please add page numbers to the table of content on the first page.

8) Please provide the source data as one zipped folder per figure.

9) During our routine image and data integrity check that we perform on all manuscript prior to acceptance, we noted the following issues that need to be resolved:

A) Cell reuse not listed in the figure legend between Figure 1A, EV1A and 7A

- The ev control and OV images are the same in all three figure panels, but rotated in Figure EV1A.

- In addition, the images shown for ulp-4 (RNAi), control and OV are the same in Figure 1A and EV Figure 1A, but again rotated in the latter.

B) Images for the ev condition from Figure 5A and 5B are shown again as L4 in Figure 7G and 7H.

C) Images shown in Figure 5E,F,G are again shown in Appendix Figure S5.

The reuse of these images must be clearly stated in the respective figure legends and is only allowed if the controls are appropriate for all these experimental perturbations and indeed derived from the same experiment. Even if this applies, I strongly recommend using this space to show different representative images from the same experiment, to showcase variation and/or reproducibility.

10) I noticed a type in the Data Availability section. It should be "would require deposition" instead of "deposit".

11) Our production/data editors have asked you to clarify several points in the figure legends (see below). Please incorporate these changes in the manuscript and return the revised file with tracked changes with your final manuscript submission.

A) Statistical test information. Only p-values that are actually shown in the figure panel(s) should (and must) be defined in the legends, all others should be removed from (or added to) the legend. Moreover, we ask for the specification of exact p-values, unless the p-values are small ($p < 0.0001$), which can be reported as inequalities:

- Please note that the exact p values are not provided in the legends of figures 1B-E; 2B, C; 3A, B, E; 5C, D; 6B, C; 7B, D, E, F, I, J; EV1 C, EV2 B, EV3 C, EV5 B, S1 B, S4B, C

B) Replicates and error bars: Please note that the error bars are not defined in the legends of figures EV5 A, B.

C) Data presentation: Please note that the white arrows are not defined in the legend of figure S5. This needs to be rectified.

12) All graphs need to show the individual data points (or individual means of each replicate) in addition to the mean and error bar.

13) Figure 4B: one of the error bars shows "2 um" in the image. Please note that the size of the error bar should only be specified in the figure legend, not in the image itself.

14) The scale bar in Figure EV1A is not well visible.

15) As a standard procedure we edit the title and abstract to make it more accessible to our general readership. Please find my suggestions below my signature. If possible, the abstract should be further shortened to approx. 175 words.

16) Finally, EMBO Reports papers are accompanied online by

A) a short (1-2 sentences) summary of the findings and their significance,

B) 2-3 bullet points highlighting key results and

C) a schematic summary figure that provides a sketch of the major findings (not a data image).

Please provide the summary figure as a separate file in PNG or JPG format at a size of 550x300-600 pixels (width x height).

Please note that the size is rather small and that text needs to be readable at the final size. Please send us this information along with the revised manuscript.

With kind regards,

=====

Referee #1:

[supports publication in the summary sheet returned with the report]

Referee #2:

The revised manuscript by Zhang and Samuelson has sufficiently addressed the reviewer comments and is suitable for publication in EMBO Reports. While I agree with Reviewer #1 that direct evidence of DRH-1 SUMOylation is a limitation of this study, the authors' justification of challenging technical limitations is sensible (particularly given the inconsistent and relatively small number of Orsay-infected cells in any given animal), making this direct analysis unlikely to be successful. In this reviewer's view, there is sufficient and rigorous indirect evidence to support the proposed model.

Minor suggestions:

Introduction - The authors should consider adjusting or expanding the range of the predicted evolutionary divergence of *C. elegans* and *C. briggsae*. See Cutter et al. 2008, PMID: 18234705

Results and Appendix Figure S1 - There is mixed terminology surrounding acute heat shock versus prolonged heat stress. Prolonged heat stress, 28-30C for ~24h, is a *pals-5p::GFP* inducer but is referred to as "heat shock" in the results text. The terminology is also mixed in Appendix Figure S1A label versus the legend text. This treatment should be consistently referred to as heat stress for clarity. By contrast, heat shock - 37C 2h - as described in the introduction related to IPR-regulated ubiquitin ligases is correct.

=====

Suggested title and abstract:

Aging impairs the antiviral defense in *Caenorhabditis elegans* due to loss of DRH-1/RIG-I deSUMOylation

Age-induced loss of DRH-1/RIG-I deSUMOylation by ULP-4/SEN7 impairs the antiviral defense in *Caenorhabditis elegans*

The innate immune defense relies on post-translational modifications (PTMs) to protect against viral infections. SUMOylation plays complex roles in viral replication and antiviral defenses in mammals and has been implicated in age-associated diseases. Whether PTMs and SUMOylation contribute to the age-induced immunosenescence, is unknown. Here, we find that the antiviral defense in *Caenorhabditis elegans* is regulated through SUMOylation of DRH-1, the ortholog of cytosolic pattern recognition receptor RIG-I, and that this regulation breaks down during aging. The SUMO isopeptidase ULP-4 is essential for deSUMOylation of DRH-1 and activation of the intracellular pathogen response (IPR) after exposure to Orsay virus (OV), a natural enteric *C. elegans* pathogen. ULP-4 stabilizes DRH-1, which translocates to the mitochondria to activate the IPR in young animals exposed to virus. Loss of either *drh-1* or *ulp-4* compromises the antiviral defense resulting in a failure to clear the virus and to protect from intestinal pathogenesis. During aging, expression of *ulp-4* decreases, resulting in increased proteasomal degradation of DRH-1 and loss of the IPR. A mutant version of DRH-1 lacking the SUMOylated lysines induces a constitutive activation of the IPR in young animals and partially rescues the IPR in aged animals. Our work establishes that aging results in dysregulated SUMOylation and loss of DRH-1, which compromises antiviral defense and creates a physiological shift to favor chronic pathological infection in older animals.

Detailed Response to the critique.

EMBOR-2024-60913V2

Browsing through the manuscript myself, I noticed a few editorial things that we need before we can proceed with the official acceptance of your study.

1) Please provide up to 5 keywords on the title page.

We have added the keywords: “*C. elegans*, Orsay virus, antiviral defense, aging, SUMOylation”.

2) Please update the 'Conflict of interest' paragraph to our new 'Disclosure and competing interests statement'. For more information see

<https://www.embopress.org/page/journal/14693178/authorguide#conflictsofinterest>

Done.

3) Please update the references to the alphabetical Harvard style. The abbreviation 'et al' should be used if there are more than 10 authors. You can download the respective EndNote file from our Guide to Authors

https://endnote.com/style_download/embo-reports/

Done.

4) The manuscript sections should be in the following order: Title page - Abstract & Keywords - Introduction - Results - Discussion - Methods - Data Availability - Acknowledgments - Disclosure Statement & Competing Interests - References - Figure Legends - (Main Tables with legends if applicable) - Expanded View Figure Legends.

Fixed.

- Materials and Methods should be just Methods

Fixed.

5) The Significance statement is not part of EMBO Reports articles and needs to be removed.

We have removed the Significance statement.

6) Please provide a legend for Dataset EV1. The legend needs to be in a separate tab of the .xls file.

As requested, we have added a legend in a separate tab to Dataset EV1 within the revised .xls file.

7) The individual Appendix Figure files are not required, the merged Appendix PDF is sufficient. But please add page numbers to the table of content on the first page.

Done.

8) Please provide the source data as one zipped folder per figure.

Done.

9) During our routine image and data integrity check that we perform on all manuscript prior to acceptance, we noted the following issues that need to be resolved:

We have updated all of the images or provide additional context to the legend. Please see details below.

A) Cell reuse not listed in the figure legend between Figure 1A, EV1A and 7A

- The ev control and OV images are the same in all three figure panels, but rotated in Figure EV1A.

- In addition, the images shown for *ulp-4* (RNAi), control and OV are the same in Figure 1A and EV Figure 1A, but again rotated in the latter.

We apologize for the confusion and oversight. In our resubmission we retained the use of the same four images in Figure 1A and EV1A: empty vector control and *ulp-4*(RNAi) treated animals, with and without OV treatment; Figure 1A highlights the conditions that are germane for rest of our study and helps focus the reader. In contrast, the larger set of genetic conditions that were tested within that same set of experiments (shown in EV1A) are both comprehensive and supportive of the general conclusion. The images are rotated for aesthetic placement of the images within each figure. In early iterations of our manuscript, we used the EV1A panels as part of the main figure, but felt that it diluted the subsequent focus to the role of ULP-4 in regulating DRH-1. We have updated both the legend for both Figure 1 and EV1 as requested to indicate that a panel is used in both.

Since Figure 1A and 7A are the control conditions for testing two distinct hypotheses, we agree that showing different representative images for the young animal controls in Figure 7A would be most appropriate. We have updated Figure 7A with new images that were also taken within the same experiment as the images of the older animals, and added these images to the source data files.

B) Images for the ev condition from Figure 5A and 5B are shown again as L4 in Figure 7G and 7H.

We have done as requested and replaced all of the representative images in Figure 5A and 5B, which were taken in a second independent trial, and added these images to the source data files.

C) Images shown in Figure 5E,F,G are again shown in Appendix Figure S5.

The images in Figure 5E,F,G are composite images from three fluorescent channels (red, green, and blue). In Appendix Figure S5, we show the individual fluorescent channels to allow the reader to clearly see each channel separately, as well as the composite image. Because they are important controls, but dilute focus, we chose to place the individual fluorescent channel images within Appendix Figure S5. We have indicated the reuse of the composite image in both legends.

The reuse of these images must be clearly stated in the respective figure legends and is only allowed if the controls are appropriate for all these experimental perturbations and indeed derived from the same experiment. Even if this applies, I strongly recommend using this space to show different representative images from the same experiment, to showcase variation and/or reproducibility.

Thank you for the clarification. In all cases we have followed these instructions.

10) I noticed a typo in the Data Availability section. It should be "would require deposition" instead of "deposit".

Thank you for catching our mistake. Fixed.

11) Our production/data editors have asked you to clarify several points in the figure legends (see below). Please incorporate these changes in the manuscript and return the revised file with tracked changes with your final manuscript submission.

A) Statistical test information. Only p-values that are actually shown in the figure panel(s) should (and must) be defined in the legends, all others should be removed from (or added to) the legend. Moreover, we ask for the specification of exact p-values, unless the p-values are small ($p < 0.0001$), which can be reported as inequalities:
- Please note that the exact p values are not provided in the legends of figures 1B-E; 2B, C; 3A, B, E; 5C, D; 6B, C; 7B, D, E, F, I, J; EV1 C, EV2 B, EV3 C, EV5 B, S1 B, S4B, C

We have done as requested and provide the exact P-values within the legend and/or the actual figure itself.

B) Replicates and error bars: Please note that the error bars are not defined in the **legends** of figures EV5 A, B.

We apologize for the accidental omission and have added this information to the legend (SEM).

C) Data presentation: Please note that the white arrows are not defined in the legend of figure S5. This needs to be rectified.

Thank you for catching this, in the revised manuscript we added to the legend that the arrows highlight DRH-1 localization to the outer membrane of the mitochondria.

12) All graphs need to show the individual data points (or individual means of each replicate) in addition to the mean and error bar.

We have updated all of the graphs to show individual data points.

13) Figure 4B: one of the error bars shows "2 μm " in the image. Please note that the size of the error bar should only be specified in the figure legend, not in the image itself.

Please note, we no longer have access to the departmental computer and the corresponding software attached to the microscope that was used in acquiring the image in question. We would have been limited to using imaging software (Photoshop) to manually remove the "2 μm " from the bottom corner of the original image and therefore preferred to keep the label.

14) The scale bar in Figure EV1A is not well visible.

We have thickened the error bar to improve visibility.

15) As a standard procedure we edit the title and abstract to make it more accessible to our general readership. Please find my suggestions below my signature. If possible, the abstract should be further shortened to approx. 175 words.

Thank you for the suggested edits to the title and abstract! We have integrated your suggestions and reworked the abstract to 175 words.

We have slightly altered the title to: “Aging impairs the antiviral defense in *Caenorhabditis elegans* due to loss of DRH-1/RIG-I deSUMOylation by ULP-4/SEN7”.

16) Finally, EMBO Reports papers are accompanied online by
A) a short (1-2 sentences) summary of the findings and their significance,

Summary/Significance:

Immunosenescence represents a shift in physiological state that broadly encompasses an age-associated decline in the ability successfully resolve pathogen infections. Using the genetically tractable roundworm *Caenorhabditis elegans*, we have discovered that dysregulated SUMOylation of the RIG-I ortholog, a key “watchtower” for viral infection, contributes to immunosenescence during aging.

B) 2-3 bullet points highlighting key results and

- **The SUMO isopeptidase ULP-4 (SEN7 ortholog) is essential for activation of the viral sensor dicer-related helicase (DRH-1; RIG-I ortholog), induction of antiviral defense, and limiting viral pathogenesis.**
- **Declining *ulp-4* expression and dysregulated SUMOylation of DRH-1 during aging compromises antiviral defense.**
- **Blocking DRH-1 SUMOylation partially preserved the antiviral response in older animals.**

C) a schematic summary figure that provides a sketch of the major findings (not a data image). Please provide the summary figure as a separate file in PNG or JPG format at a size of 550x300-600 pixels (width x height). Please note that the size is rather small and that text needs to be readable at the final size. Please send us this information along with the revised manuscript.

We have provided a schematic with our resubmission.

With kind regards,

=====

Referee #1:
[supports publication in the summary sheet returned with the report]

Referee #2:

The revised manuscript by Zhang and Samuelson has sufficiently addressed the reviewer comments and is

suitable for publication in EMBO Reports. While I agree with Reviewer #1 that direct evidence of DRH-1 SUMOylation is a limitation of this study, the authors' justification of challenging technical limitations is sensible (particularly given the inconsistent and relatively small number of Orsay-infected cells in any given animal), making this direct analysis unlikely to be successful. In this reviewer's view, there is sufficient and rigorous indirect evidence to support the proposed model.

Minor suggestions:

Introduction - The authors should consider adjusting or expanding the range of the predicted evolutionary divergence of *C. elegans* and *C. briggsae*. See Cutter et al. 2008, PMID: 18234705

We thank the reviewer for their insight! We have made the suggested adjustment to our introduction.

Results and Appendix Figure S1 - There is mixed terminology surrounding acute heat shock versus prolonged heat stress. Prolonged heat stress, 28-30C for ~24h, is a pals-5p::GFP inducer but is referred to as "heat shock" in the results text. The terminology is also mixed in Appendix Figure S1A label versus the legend text. This treatment should be consistently referred to as heat stress for clarity. By contrast, heat shock - 37C 2h - as described in the introduction related to IPR-regulated ubiquitin ligases is correct.

We thank the reviewer and have updated the text to be more precise.

=====

Suggested title and abstract:

Aging impairs the antiviral defense in *Caenorhabditis elegans* due to loss of DRH-1/RIG-I deSUMOylation

Age-induced loss of DRH-1/RIG-I deSUMOylation by ULP-4/SEN7 impairs the antiviral defense in *Caenorhabditis elegans*

The innate immune defense relies on post-translational modifications (PTMs) to protect against viral infections. SUMOylation plays complex roles in viral replication and antiviral defenses in mammals and has been implicated in age-associated diseases. Whether PTMs and SUMOylation contribute to the age-induced immunosenescence, is unknown. Here, we find that the antiviral defense in *Caenorhabditis elegans* is regulated through SUMOylation of DRH-1, the ortholog of cytosolic pattern recognition receptor RIG-I, and that this regulation breaks down during aging. The SUMO isopeptidase ULP-4 is essential for deSUMOylation of DRH-1 and activation of the intracellular pathogen response (IPR) after exposure to Orsay virus (OV), a natural enteric *C. elegans* pathogen. ULP-4 stabilizes DRH-1, which translocates to the mitochondria to activate the IPR in young animals exposed to virus. Loss of either *drh-1* or *ulp-4* compromises the antiviral defense resulting in a failure to clear the virus and to protect from intestinal pathogenesis. During aging, expression of *ulp-4* decreases, resulting in increased proteasomal degradation of DRH-1 and loss of the IPR. A mutant version of DRH-1 lacking the SUMOylated lysines induces a constitutive activation of the IPR in young animals and partially rescues the IPR in aged animals. Our work establishes that aging results in dysregulated SUMOylation and loss of DRH-1, which compromises antiviral defense and creates a physiological shift to favor chronic pathological infection in older animals.

Dr. Andrew Samuelson
University of Rochester Medical Center
Biomedical Genetics
United States

Dear Andrew,

Thank you for implementing the final minor changes to the figures. I am now very pleased to accept your manuscript for publication in the next available issue of EMBO reports. Thank you for your contribution to our journal.

Kind regards,
